# On the conversational persuasiveness of GPT-4

Francesco Salvi [1,2] ✉, Manoel Horta Ribeiro [3], Riccardo Gallotti [2] &
Robert West [1]

Early work has found that large language models (LLMs) can generate
persuasive content. However, evidence on whether they can also personalize
arguments to individual attributes remains limited, despite being crucial for
assessing misuse. This preregistered study examines AI-driven persuasion
in a controlled setting, where participants engaged in short multiround
debates. Participants were randomly assigned to 1 of 12 conditions in a
2 × 2 × 3 design: (1) human or GPT-4 debate opponent; (2) opponent with
or without access to sociodemographic participant data; (3) debate topic
of low, medium or high opinion strength. In debate pairs where AI and
humans were not equally persuasive, GPT-4 with personalization was more
persuasive 64.4% of the time (81.2% relative increase in odds of higher
post-debate agreement; 95% confidence interval [+26.0%, +160.7%], $P < 0.01$;
$N = 900$). Our findings highlight the power of LLM-based persuasion and
have implications for the governance and design of online platforms.

Persuasion, the process of altering someone's belief, position or opinion on a specific matter, is pervasive in human affairs and a widely studied topic in the social sciences[1-3]. From public health campaigns[4-6] to marketing and sales[7,8] to political propaganda[9,10], various actors develop elaborate persuasive communication strategies on a large scale, investing substantial resources to make their messaging resonate with broad audiences. In recent decades, the diffusion of social media and other online platforms has expanded the potential of mass persuasion by enabling personalization or 'microtargeting'—the tailoring of messages to an individual or a group to enhance their persuasiveness[11,12]. The efficacy of microtargeting has been questioned because it relies on the assumption of effect heterogeneity, that is, that specific groups of people respond differently to the same inputs, a concept that has been disputed in previous literature[13,14]. Nevertheless, microtargeting has proven effective in a variety of settings[15-17], and most scholars agree on its persuasive power[15,18,19].

Microtargeting practices are fundamentally constrained by the burden of profiling individuals and crafting personalized messages that appeal to specific targets, as well as by a restrictive interaction context without dialogue. These limitations may soon fall off due to the recent rise of large language models (LLMs)—machine learning models trained to mimic human language and reasoning by ingesting vast amounts of textual data. Models such as GPT-4 (ref. 20), Claude[21] and Gemini[22] can generate coherent and contextually relevant text with fluency and versatility, and exhibit human or superhuman performance in a wide range of tasks[23]. In the context of persuasion, experts have widely expressed concerns about the risk of LLMs being used to manipulate online conversations and pollute the information ecosystem by spreading misinformation, exacerbating political polarization, reinforcing echo chambers and persuading individuals to adopt new beliefs[24-27].

A particularly menacing aspect of AI-driven persuasion is its possibility to easily and cheaply implement personalization, conditioning the models' generations on personal attributes and psychological profiles[28]. This is especially relevant since LLMs and other AI systems are capable of inferring personal attributes from publicly available digital traces such as Facebook likes[29,30], status updates[31,32] and messages[33], Reddit and Twitter posts[34,35], pictures liked on Flickr[36], and other digital footprints[37]. In addition, users find it increasingly challenging to distinguish AI-generated from human-generated content, with LLMs efficiently mimicking human writing and thus gaining credibility[38-41].

Recent work has explored the potential of AI-powered persuasion by comparing texts authored by humans and LLMs, finding that modern language models can generate content perceived as at least on par with, and often more persuasive than, human-written content[41-46].

[1]EPFL, Lausanne, Switzerland. [2]Fondazione Bruno Kessler, Trento, Italy. [3]Princeton University, Princeton, NJ, USA. ✉e-mail: francesco.salvi@epfl.ch

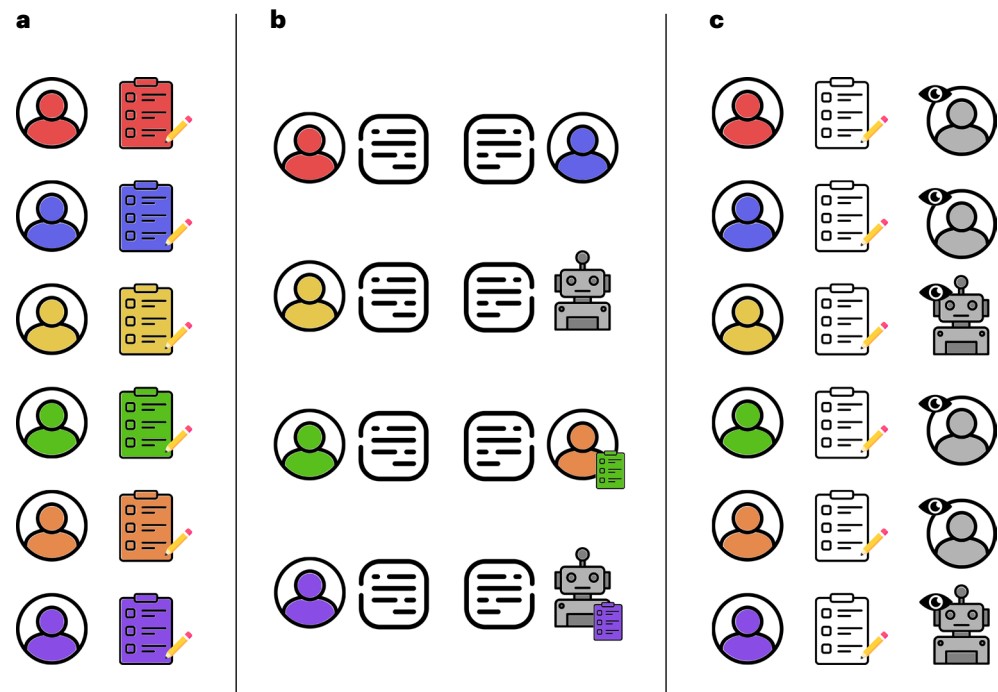

**Fig. 1 | Overview of the experimental design. a**, Participants complete a sociodemographic survey (gender, age, ethnicity, education level, employment status, political affiliation). **b**, Every 5 min, participants who have completed the survey are randomly assigned to one of four treatment conditions: Human–Human, Human–AI, Human–Human (personalized) and Human–AI (personalized). In the 'personalized' conditions, the opponent can access information collected from the participant's survey. Participant and opponent then debate for 10 min on a randomly assigned topic, holding the PRO or CON standpoint as instructed. Topics are randomly drawn from sets of three levels of opinion strength (low, medium, high). **c**, After the debate, participants complete another short survey measuring opinion change. Finally, they are debriefed about their opponent's identity. Our key outcome is the change in participants' views on the debated topic from before versus after the debate.

Other research has focused on personalization, observing consequential yet non-unanimous evidence about the impact of LLMs on microtargeting[47–49]. There is, however, still limited knowledge about the persuasive power of LLMs in direct conversations with human counterparts and how AI persuasiveness, with or without personalization, compares with human persuasiveness (see Supplementary Section 1 for an additional literature review). We argue that the direct-conversation setting is of particularly high practical importance, as commercial LLMs such as ChatGPT, Claude and Gemini are trained for conversational use[50].

In this preregistered study, we examine the effect of AI-driven persuasion in a controlled, direct-conversation setting. We created a web-based platform where participants engage in short multiround debates on various sociopolitical issues. Each participant was randomly paired with either GPT-4 or a live human opponent and assigned to a topic and a stance to hold. To study the effect of personalization, we also experimented with a condition where opponents had access to sociodemographic information about participants, thus granting them the possibility of tailoring their arguments to individual profiles. In addition, we experimented with three sets of debate topics, clustered on the basis of the strength of participants' previous opinions. The result is a 2 × 2 × 3 factorial design (two opponent types, two levels of participant information, three levels of topic strength). By comparing participants' agreement with the debate proposition before versus after the debate, we can measure shifts in opinion and, consequently, compare the persuasive effect of different treatments. Our setup differs substantially from previous research in that it enables a direct comparison of the persuasive capabilities of humans and LLMs in real conversations, providing a framework for benchmarking how state-of-the-art models perform in online environments and the extent to which they can exploit personal data. Although our study used a

structured debate format, it nonetheless serves as a valuable proof of concept for how similar debates occur online, such as in synchronous discussions on platforms such as Facebook and Reddit.

## Experimental design

Participants ($N = 900$) recruited for our experiment were redirected to a custom-made web platform designed to support real-time interactive conversations (the platform was built on top of Empirica.ly[51]; see Supplementary Section 2.4 for details). The experiment's workflow is represented schematically in Fig. 1. In phase A, participants asynchronously completed introductory steps and filled in a short demographic survey, recording their gender, age, ethnicity, education level, employment status and political affiliation. At each clock trigger of a 5-min interval timer, all participants who had completed the survey were randomly assigned to a treatment condition and matched with an appropriate opponent. In addition, each participant–opponent pair was randomly assigned to one debate topic (a simple debate proposition, for example, "Should students have to wear school uniforms?"; see Supplementary Section 2.3 for the complete list) and a random role, either in support (PRO) or against (CON) the proposition.

After being matched, participants transitioned to phase B, which was in turn divided into four stages: (1) 'screening' (1 min), where participants, without yet knowing their role, were asked how much they agreed with the debate proposition (on a scale from 1, 'strongly disagree', to 5, 'strongly agree') and how much they had previously thought about it; (2) 'opening' (4 min), where participants articulated their main arguments coherently with the assigned role; (3) 'rebuttal' (3 min), where they responded to their opponent's arguments; and (4) 'conclusion' (3 min), where participants were asked to respond to their opponent's rebuttal or reiterate their initial points. The opening–rebuttal–conclusion structure is based on a simplified version of the format

commonly used in competitive academic debates. After the debate, in phase C, participants asynchronously completed a final exit survey. They were asked again how much they agreed with the proposition and whether they believed their opponent was a human or an AI. Finally, they were debriefed about their opponent's true identity.

For each debate, the assigned topic was randomly drawn from 30 propositions encompassing salient US political and social issues. An online sample of 60 US residents manually validated and selected these propositions from a larger pool, recording also their agreement with each proposition. We divided topics into three clusters (low, medium, high) of 10 propositions each, on the basis of their average opinion strength (see Supplementary Section 2 for details and the full list of topics). In this context, 'strength' intuitively reflects a topic's level of controversy or polarization and how easy it is to sway opinions about it. Each participant was assigned to 1 of 12 conditions in a 2 × 2 × 3 design:

- The participant was assigned to debate either a human or a GPT-4 opponent.
- The opponent (whether human or GPT-4) either did or did not receive sociodemographic information (gender, age, ethnicity, education level, employment status and political affiliation, as recorded in the pretreatment survey) about the participant. When the LLM was provided with personal attributes, we instructed it to 'astutely use this information to craft arguments that are more likely to persuade and convince your opponent of your stance' (see Supplementary Section 2.5 for the complete prompts).
- The debate topic belonged to the low-, medium- or high-strength cluster.

In each condition, 50 debates were held, yielding 600 debates and $N = 900$ unique participants, since Human–Human debates required two participants. Within each condition, we randomized topics such that exactly 5 debates argued about each topic.

We measured the persuasive effect of the treatment conditions described by measuring participants' agreements with their propositions before ($A^{\mathrm{pre}}$) and after ($A^{\mathrm{post}}$) the debates. To frame changes in agreement as persuasive effects, we aligned the scores with the side (PRO or CON) 'opposed' to the one assigned to the participant, that is, the one held by their opponent, by transforming them as follows:

$$\tilde{A} = \begin{cases} 6 - A & \text{if participant side = PRO,} \\ A & \text{if participant side = CON,} \end{cases} \quad (1)$$

resulting in the two variables $\tilde{A}^{\mathrm{pre}}$ and $\tilde{A}^{\mathrm{post}}$. Implicitly, this transformation corresponds to the natural assumption that agreements get inverted around 3 (the 'neutral' score) when debate propositions are negated. With this adjustment, $\tilde{A}^{\mathrm{post}} > \tilde{A}^{\mathrm{post}}$ means that participants were persuaded to shift their opinion towards their opponent's side, whereas $\tilde{A}^{\mathrm{post}} \leq \tilde{A}^{\mathrm{post}}$ means that their opinion did not change or was reinforced towards their assigned side. By comparing the transformed agreement scores $\tilde{A}^{\mathrm{post}}$ and $\tilde{A}^{\mathrm{post}}$ using a partial proportional odds model[52], we measure the causal effect of each treatment condition on the likelihood that participants are persuaded by their opponents. In particular, we consider as our main outcome the odds $\frac{P(\tilde{A}^{\mathrm{post}} > a)}{P(\tilde{A}^{\mathrm{post}} \leq a)}$ of obtaining higher post-treatment agreement, $\forall\, a \in \{1, 2, 3, 4\}$. We chose this model because the outcome is ordinal and since our data do not satisfy assumptions of simpler ordinal regression models (see Supplementary Section 3 for details).

## Results
### Aggregate results
Our key finding is that GPT-4 performs as well as or better than humans in our debate task. We consider the first two dimensions of our design

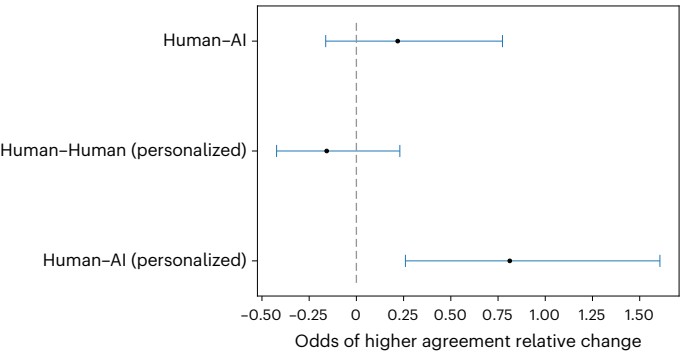

**Fig. 2 | Regression results for the partial proportional odds model.** For each condition, the point estimates represent relative changes compared to the Human–Human reference in the odds of post-treatment agreement assuming higher values (see Supplementary Section 3 for more details). Horizontal lines indicate 95% CIs based on two-sided $t$-tests; $n = 750$. GPT-4 outperforms humans in the debate task when given participants' basic personal information ($P < 0.01$) and performs similarly to humans when not given personal information. Full numerical results, including intercepts, are reported in Supplementary Table 4.

(human or AI opponents, with or without personalization), aggregating across all topic clusters. We report the results in Fig. 2, taking as a reference the Human–Human condition and examining the differences relative to it. Human–AI (personalized) debates show the strongest positive effect, meaning that GPT-4 with access to personal information had higher persuasive power than humans. We estimate that the odds of greater agreement with opponents is +81.2% (95% confidence interval (CI) [+26.0%, +160.7%], $P < 0.01$) higher in the Human–AI (personalized) condition compared with the Human–Human reference condition. Intuitively, this means that 64.4% of the time, personalized LLM debaters were more persuasive than humans, given that they were not equally persuasive (see Supplementary Section 3 for an explanation). For the Human–AI (+21.9%, 95% CI [−16.2%, +77.3%], $P = 0.30$) and the Human–Human (personalized) (−15.7%, 95% CI [−42.2%, +23.0%], $P = 0.38$) conditions, there is insufficient evidence to conclude a difference in persuasiveness between them and the Human–Human baseline, considering a 0.05 significance level. By contrast, the Human–AI (personalized) effect remains significant even when changing the reference category to Human–AI ($P = 0.04$). Remarkably, these findings provide evidence that GPT-4-based microtargeting strongly outperforms both non-personalized GPT-4 and human-based microtargeting, with GPT-4 leveraging personal information more effectively than humans (see Supplementary Section 5 for examples of complete debates showcasing effective use of personalization).

To provide a more intuitive interpretation, we repeat our analysis using a simpler linear regression (see Supplementary Section 7 for details), a modelling choice that is statistically less appropriate for our setup (see Methods for a broader discussion) but that can intuitively provide some basic insights about the effectiveness of our treatments. We find that, on average, Human–AI (personalized) debates are associated with an increase of 0.36 (95% CI [0.12, 0.60], $P < 0.01$) in the difference $\tilde{A}^{\mathrm{post}} - \tilde{A}^{\mathrm{post}}$, with respect to the Human–Human condition. In contrast, the other two treatments again have non-significant coefficients.

### Absolute changes
Complementing the results concerning relative change, we also inspect the absolute agreement distributions (see Supplementary Section 8 for details). We find that debates tended to produce a backfire reaction for all conditions except Human–AI (personalized), reinforcing opinions toward the side assigned for the experiment instead of moving them toward the opponent's side. This trend is consistent with previous literature describing a hardening of pretreatment opinions when people

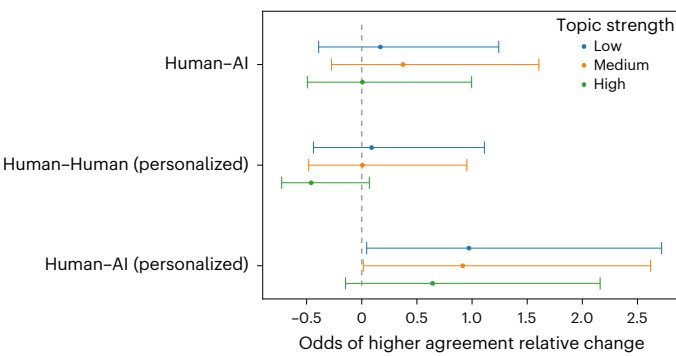

**Fig. 3 | Regression results for the partial proportional odds model, controlling for topic strength.** For each condition, the point estimates represent relative changes compared to the Human–Human reference in the odds of post-treatment agreements assuming higher values. Horizontal lines indicate 95% CIs based on two-sided *t*-tests; *n* = 750. Full numerical results are reported in Supplementary Tables 5, 7.

express their ideas[53] or are exposed to disagreeing views[54], or finding opinion change to be highly affected by argument order[55].

## Topic clusters
Figure 3 shows the results broken down by topic strength, fitting an independent regression per cluster. We observe that the effect of Human–AI (personalized) debates remains strong for the low- and medium-strength clusters, but drops below the significance level for high-strength topics (+64.2%, 95% CI [−14.7%, +216.1%], *P* = 0.14). Again, the effects across all clusters remain non-significant for the other conditions, exhibiting persuasive results indistinguishable from the Human–Human baseline.

## Linguistic patterns
Next, we investigate how arguments differ across treatment conditions by conducting a textual analysis of the generated writings to identify distinctive patterns. In Fig. 4, we report the distribution of prominent textual features extracted with LIWC-22 (ref. 56) (see Supplementary Section 10 for additional details). We observe that GPT-4 opponents tended to use logical and analytical thinking substantially more than humans. On the other hand, humans used more first-person singular and second-person pronouns and produced longer but easier-to-read texts, as measured by the Flesch Reading Ease score[57]. The difference in length and second-person pronoun usage can, at least partially, be explained by the specific prompts we chose (see Supplementary Section 2.5 for details), where we instructed GPT-4 to write only one to two sentences per stage and to refrain from directly addressing its opponent unless they do it first. There does not seem to be a difference induced by personalization, with distributions being very similar both between Human–Human and Human–Human (personalized) and between Human–AI and Human–AI (personalized). Analyses of underlying social dimensions and usage of persuasive strategies (Supplementary Section 10) confirm that GPT-4 heavily relied on logical reasoning and factual knowledge. At the same time, humans displayed more appeals to similarity, expressions of support and trust, and employed more storytelling.

## Perceived opponent
Finally, we turn to participants' perceptions of their opponents, recorded at the end of each debate by asking them whether they thought they had debated with a human or an AI. Figure 5 shows the distribution of answers and how the difference in pre- versus post-debate agreement ($\bar{A}^{post} - \bar{A}^{post}$) depends on how participants perceived their opponents. In debates with AI, participants correctly identified their opponent's identity in about three out of four cases,

indicating that the writing style of GPT-4 in this setting has distinctive features that are easy to spot. Conversely, participants struggled to identify their opponents in debates with other humans, with a success rate indistinguishable from random chance (*P* = 0.42 for a two-sided Binomial test; success rate 52.0%, 95% CI [47.3%, 56.7%], *n* = 450). Moreover, we notice that when participants believed they were debating with an AI, they changed their expressed scores to agree more with their opponents compared with when they believed they were debating with a human (odds of greater agreement with opponents +37.4%, 95% CI [+3.03%, +83.3%], *P* = 0.03; see Supplementary Section 12 for details). We emphasize that this observation is based solely on correlation and does not imply causation: it is unclear whether the difference in agreement change is motivated by participants' beliefs about their opponent or whether, conversely, those beliefs are caused by opinion change. For example, participants could have been more lenient in changing their agreement score towards their opponent when they believed they were facing an AI, because not having a human on the other side makes it unconsciously easier to accept that they have somewhat lost the debate. Conversely, participants could have also believed that their opponent was an AI because of how well their arguments were written. However, even if differences in agreement change were influenced by how participants perceived their opponent (in line with the first of the above two explanations), we find that treatment effects change very little when adding beliefs about opponents' identities as a control in our regression. Particularly, the Human–AI (personalized) condition still has a strong and significant effect (+70.2%, 95% CI [+17.8%, +146.0%], *P* < 0.01). Therefore, how participants perceived their opponent is not enough to explain the treatment effects, which instead seem more tied to the intrinsic capabilities of AI to generate better arguments. Lastly, we investigate the relationship between perceptions of opponents and textual covariates, finding that participants associated texts that are easy to read (*P* = 0.04) with human opponents (see Supplementary Section 12 for details).

## Discussion
LLMs have been criticized for their potential to generate and foster the diffusion of hate speech, misinformation and malicious political propaganda. Specifically, there are concerns about the persuasive capabilities of LLMs, which could be critically enhanced through personalization, that is, tailoring content to individual targets by crafting messages that resonate with their specific background and demographics[25,26,28].

In this paper, we explored the effect of AI-driven persuasion and personalization in structured online conversations, comparing the performance of GPT-4 with that of humans in a one-on-one debate task. We conducted a controlled experiment where we assigned participants to 1 of 12 treatment conditions, randomizing their debate opponent to be either a human or GPT-4, as well as randomizing access to personal information and the degree of opinion strength of the debate topic. We then compared reported agreements before and after the debates, measuring the opinion shifts of participants and, thus, the persuasive power of the arguments generated by humans and AI.

Our results show that, on average, GPT-4 opponents outperformed human opponents across every topic and demographic, exhibiting a high level of persuasiveness. In particular, when compared to the baseline condition of debating with a human, debating with GPT-4 with personalization resulted in a +81.2% increase (95% CI [+26.0%, +160.7%], *P* < 0.01) in the odds of reporting higher agreements with opponents (see Supplementary Section 3 for details). More intuitively, this means that 64.4% of the time, personalized GPT-4 opponents were more persuasive than human opponents, given that they were not equally persuasive (see Supplementary Section 3 for a detailed explanation). Without personalization, GPT-4 opponents were on par with human opponents (*P* = 0.30), and so were human opponents with access to personalization (*P* = 0.38). In other words, not only was GPT-4 able to

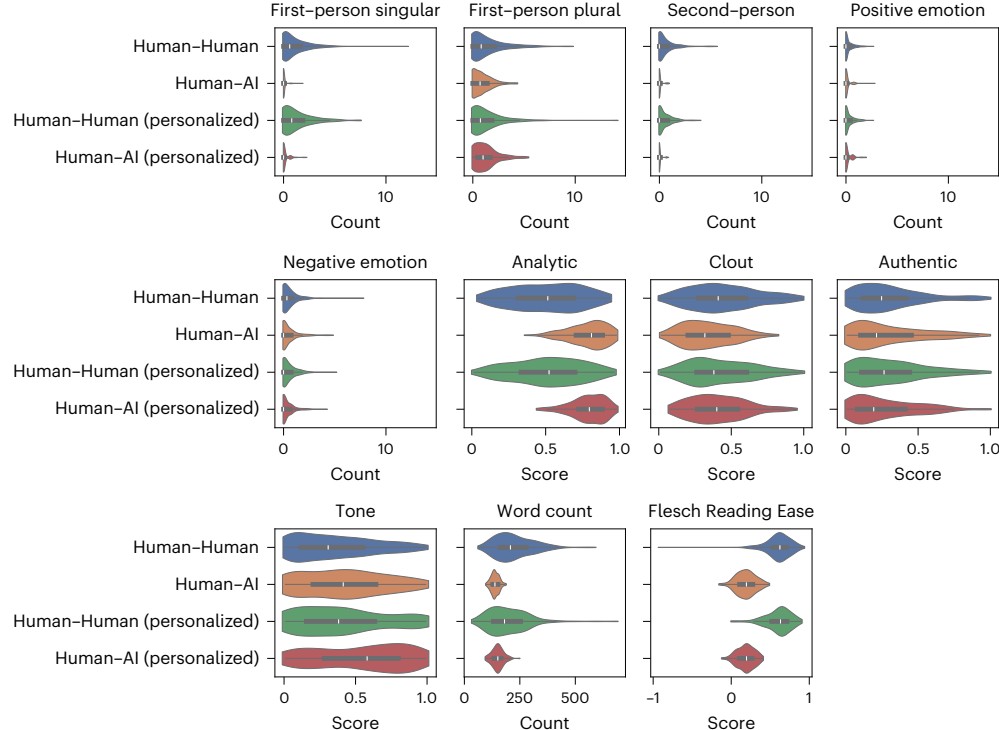

**Fig. 4 | Distribution of textual features by treatment condition.** Each violin is drawn using a kernel density estimate of the underlying distribution. Within each violin, a box plot is overlaid: the centre line indicates the median, the bounds of the box represent the first and third quartiles (Q1 and Q3), and the whiskers extend to the minimum and maximum values within 1.5 times the interquartile range from Q1 to Q3. Points beyond this range are omitted; $n = 750$. Except for the Flesch Reading Ease score, all features were extracted via LIWC-22 (ref. 56), which provides a dictionary of words belonging to various linguistic, psychological and topical categories (see Supplementary Section 10 for additional details). 'Analytic' is a metric of logical, formal and analytical thinking, 'Clout' expresses language of leadership and status, 'Authentic' measures perceived honesty and genuineness, and 'Tone' is the degree of emotional tone. Flesch Reading Ease is a measure of how easy to read a text is, based on its average number of words per sentence and average number of syllables per word. Analytic, Clout, Authentic and Tone have been normalized to the [0, 1] range, Flesch Reading Ease scores were divided by 100, while the remaining categories were computed directly as frequencies across the entire text produced by each participant. Note that scales differ across panels representing counts and scores, and additionally for the Word count and the Flesh Reading Ease panels.

exploit personal information to tailor its arguments effectively, but it also succeeded in doing so far more effectively than humans.

Our study suggests that concerns around personalization and AI persuasion are warranted, reinforcing previous results[42,44,45,48] by showcasing how LLMs can outpersuade humans in online conversations through microtargeting. We emphasize that the effect of personalization is particularly remarkable given how little personal information was collected (gender, age, ethnicity, education level, employment status and political affiliation) and despite the extreme simplicity of the prompt instructing the LLM to incorporate such information (see Supplementary Section 2.5 for the complete prompts). Even stronger effects could probably be obtained by exploiting individual psychological attributes, such as personality traits and moral bases, or by developing stronger prompts through prompt engineering, fine-tuning or specific domain expertise. In this context, malicious actors interested in deploying chatbots for large-scale disinformation campaigns could leverage fine-grained digital traces and behavioural data, building sophisticated, persuasive machines capable of adapting to individual targets. We argue that online platforms and social media should seriously consider such threats and extend their efforts to implement measures countering the spread of AI-driven persuasion. A promising approach to counter mass disinformation campaigns could be enabled by LLMs themselves, generating similarly personalized counternarratives to educate bystanders potentially vulnerable to deceptive posts[27,58]. Early efforts in this direction are already underway, with promising results in reducing conspiratorial beliefs thanks to dialogues with GPT-4 (ref. 59). Our analyses also provided

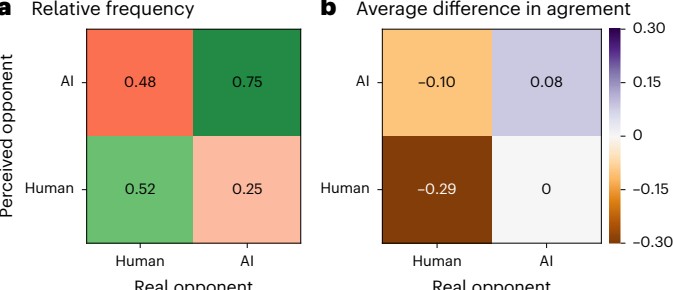

**Fig. 5 | Statistics regarding participants' perceptions of debate opponents. a**, Relative frequency of people's beliefs on whether they were debating with a human or an AI, by the real type of their opponent. **b**, Average difference in agreements after versus before the debates.

initial insights into the mechanisms behind LLM persuasion. We found notable differences in the writing style between GPT-4 and human debaters. For instance, texts generated by LLM debaters were harder to read and had more markers associated with logical and analytical reasoning (Fig. 4).

Future work could replicate our approach to benchmark the persuasive capabilities of LLMs continuously, measuring the effect of different models and prompts and their evolution over time. Moreover, our method could be extended to other settings, such as negotiation games[60] and open-ended conflict resolution, mimicking the structure

of online interactions and conversations more closely. Other efforts could explore whether our results are robust to deanonymization, measuring what happens when participants are initially informed about their opponent's identity. A crucial point that also needs further investigation is why Human–AI (personalized) debates were significantly more effective than Human–AI debates, given that our descriptive analyses of textual features have found no meaningful variations between the two conditions. We hypothesize that this difference is not driven by changes in the writing style but rather by differences in the choice of issues brought up during the debates (see Supplementary Section 5 for an example of how this mechanism might unroll). However, the precise nature of the dynamics behind this process is still a very open question. In addition, we emphasize that prompting plays a big role in the textual signature exhibited by LLMs. Future work could experiment with prompts that instruct GPT-4 to be less reliant on logical reasoning and showcase more appeals to support and trust, mimicking the style of human debaters and potentially enhancing its persuasive capabilities.

Although we believe our contribution constitutes a meaningful advance for studying the persuasive capabilities of language models, we identify four key limitations.

First, the assignment of participants to debate sides was randomized, regardless of their previous opinions on the topic. This was crucial to identify causal effects under the logistical challenge of matching participants in real time. Still, it could have introduced substantial bias in that human arguments might have been weaker than those of LLMs simply because opponents did not honestly believe in the standpoint they were advocating for. To address such concerns, we fit a version of our model that considers opponents' pretreatment agreements as a control (Supplementary Section 13). We found the effect of opponents' agreements to be non-significant ($P = 0.22$), suggesting that our results might be robust to this limitation. In addition, we repeated our main analysis restricting our dataset to opponents arguing for a standpoint aligned with their previous opinions, finding again a strong and statistically significant effect for the Human–AI (personalized) condition (+122.8%, 95% CI [+6.2%, +367.3%], $P = 0.03$). Remarkably, this seems to suggest that people can play the role assigned to them with great credibility, even when defending positions they do not spontaneously agree with. Nevertheless, given the small size of this restricted sample, future work would be needed to validate our findings while enforcing matches between people on opposing sides of each issue.

Second, our experimental design forced conversations to have a predetermined structure, strictly following the stages and rules of a debate. While we believe that our setup captures the essence of many online interactions, where people reply to each other in an almost synchronous fashion or react to others' comments in real time, it still targets an artificial environment that can substantially diverge from the dynamics of online conversations, which evolve spontaneously and unpredictably. In addition, on our platform, conversations were entirely anonymized, making them different from the normal conditions under which humans interact. Therefore, we acknowledge that the ecological validity of our findings is limited, as it is unclear how our results would generalize to natural discussions on social networks and other online platforms. For this reason, our work should be seen as providing a proof of concept about LLMs' persuasive capabilities rather than a realistic evaluation of their persuasiveness in the wild, which remains an open question for future research.

Third, the time constraint implemented in each debate stage potentially limited participants' creativity and persuasiveness, decreasing their performance. This can be especially true for the Human–Human (personalized) condition, where the participants provided with personal information about their opponents had to process and implement it without any time facilitation.

Fourth, our experiment engaged human participants recruited through Prolific, who received financial incentives for completing debates and were aware of being in a controlled experimental environment. Although previous research has found Prolific to have the best data quality among competitors and research done using Prolific to be often generalizable[61–63], the pool of workers active on the platform still differs in their sociodemographic distribution from both the overall US population and the user base of other online platforms and social media. Therefore, future work is needed to understand whether our findings can be reproduced using a more representative sample that accurately mimics the overall spectrum of human persuasive skills. In addition, it would be interesting to include human experts in our comparison, such as individuals involved in competitive debating, political campaigns or public communication.

Despite these limitations, we hope our work will stimulate researchers and online platforms to seriously consider the threat posed by LLMs fuelling divide, spreading malicious propaganda and developing adequate countermeasures.

## Methods

### Human sample

Our platform was approved by EPFL's Human Research Ethics Committee (095-2023) and preregistered at https://aspredicted.org/DCC_NTP on 18 December 2023. Informed consent was collected from all participants. We recruited participants for our study through Prolific between December 2023 and April 2024, under the criteria that they were 18+ years old and located in the United States. The location requirement is motivated by the fact that most debate topics are deeply rooted in US national issues and would not resonate with different populations. To prevent skill disparity, each worker was allowed to only participate in one debate. The participant was paid €2.50 (US$3.15) and had a median completion time of 16 min, corresponding to a pay rate of about €9.40 per hour (US$11.80 per hour). Following recommendations from ref. 64, workers were explicitly informed that using LLMs and Generative AI tools was strictly prohibited and would result in their exclusion from the study. Regardless, coherent with our preregistration, we manually reviewed each debate and excluded 20 debates where at least one human participant showed clear indications of LLM usage (unrealistic values of words per minute, blatant evidence of ChatGPT's standard writing style) and plagiarism (as detected by DupliChecker, https://www.duplichecker.com/). In addition, we excluded 13 debates where at least one participant provided unacceptable (empty texts, nonsensical or few-word arguments) or incomplete answers. The number of people involved in rejected debates was not counted towards the total number of participants ($N = 900$), as the affected tasks were republished on Prolific and completed by other workers. In addition, to prevent participants from attempting a Turing test, we informed them that their goal was not to spot whether their opponent was a human or an AI but rather to be as persuasive as possible during the debate. No statistical methods were used to predetermine the total number of participants, but our sample size is similar to those reported in previous publications[41,45,48,49]. Our final sample ($N = 900$) was 49.6% male, 47.7% female, 2.7% other, with the following age distribution: 11.3% 18–24 years old, 34.1% 25–34, 23.7% 35–44, 17.3% 45–54, 8.7% 55–64, 4.8% 65+. Each participant was randomly assigned with equal probability to one treatment condition and one topic.

### Topic selection

We selected the 30 topics used within our experiment using a three-step procedure (see Supplementary Section 2 for additional details): (1) we manually curated an initial pool of 60 candidate topics, drawing from various online sources under the criteria that propositions should be broad, easy to understand and to debate, and reasonably divisive. (2) We conducted a survey on Amazon Mechanical Turk, where $N = 60$ US residents annotated candidate propositions across three dimensions: agreement, knowledge and debatableness. (3) We filtered out the 10 topics with the most unanimous positions and the

remaining 20 least debatable topics, narrowing down the pool to the final 30 topics. On the basis of the strength of 'agreements' (the absolute deviation from the 'Neutral' score), we divided those topics into three clusters (low-strength, moderate-strength and high-strength).

## Regression model

For all the regressions reported in the main text, we used a partial proportional odds specification[52] to model the agreements post treatment in terms of agreements pre treatment and treatment conditions (see Supplementary Section 3 for additional details). This modelling choice was motivated by the fact that our outcome of interest, answers on a 1–5 Likert scale, is ordinal. Previous research has advised against using 'metric' models such as linear regression for ordinal data, as the practice can lead to systematic errors[65]. For example, the response categories of an ordinal variable may not be equidistant—an assumption that is required in statistical models of metric responses[66]. A solution to this issue is the use of so-called cumulative ordinal models that assume that the observed ordinal variable comes from the categorization of a latent, non-observable continuous variable[66], such as the partial proportional odds model[52]. We fit our debate dataset to such a model using a Broyden-Fletcher-Goldfarb-Shanno solver. For Human–Human personalized debates, we only considered participants who did not have access to their opponents' personal information, so that the setup is equivalent to Human–AI personalized debates. Instead, we extracted two data points from each Human–Human debate, corresponding to both participants. We computed standard errors using a cluster-robust estimator[67] to adjust for interdebate correlations. Data collection and analysis were not performed blind to the conditions of the experiments.

## Deviations from preregistration

We indicated in our preregistration (https://aspredicted.org/DCC_NTP) that we would assign participants to 1 of 9 treatment conditions, resulting from the combination of three opponent-related conditions (Human–Human, Human–AI, Human–AI personalized) and three topic-related conditions (low-, moderate- and high-strength cluster), in a 3 × 3 factorial design. We additionally registered that we might carry out a Human–Human personalized condition on a sample of topics, conditional on resource availability. In the end, we decided to run the Human–Human personalized across all topics, with the same sample size and number of debates as the other conditions. Therefore, to simplify the main text's explanation, we reformulated it by framing it as a 2 × 2 × 3 design. For completeness, we report our originally planned 3 × 3 analysis in Supplementary Section 4. We found results to be consistent with those reported in the main text (cf. Figs. 2 and 3), with Human–AI personalized having a strong and statistically significant effect across all topic clusters.

## Reporting summary

Further information on research design is available in the Nature Portfolio Reporting Summary linked to this article.

## Data availability

The debate dataset collected for our study is publicly available at https://huggingface.co/datasets/frasalvi/debategpt (ref. 68).

## Code availability

The code to fully reproduce the analyses described in this work is available on GitHub at https://github.com/epfl-dlab/debategpt (ref. 69). Data collection was performed using Empirica v.1.9.5. The study was conducted using Python 3.11, R 4.3.1 and LIWC-22.

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

## Acknowledgements

R.W.'s lab is partly supported by grants from the Swiss National Science Foundation (200021_185043, TMSGI2_211379) and H2020 (952215), and by gifts from Google and Microsoft. R.G. acknowledges the financial support received from the European Union's Horizon Europe research and innovation programme under grant agreement no. 101070190, and from the PNRR ICSC National Research Centre for High Performance Computing, Big Data and Quantum Computing (CN00000013), under the NRRP MUR programme funded by the NextGenerationEU. The funders had no role in study design, data collection and analysis, decision to publish or preparation of the manuscript.

## Author contributions

F.S., M.H.R., R.G. and R.W. designed the research. F.S. developed the debate platform and collected the data. F.S. and M.H.R. analysed the data. F.S., M.H.R., R.G. and R.W. wrote the paper.

## Funding

## Competing interests

R.W. is a Visiting Researcher at Microsoft Research and has received funding from Google and Microsoft. Those entities had no role in study design, data collection and analysis, decision to publish or preparation of the manuscript. The views and conclusions contained herein are those of the authors and should not be interpreted as representing the official policies, either expressed or implied, of the aforementioned entities. The other authors declare no competing interests.

## Additional information

**Correspondence and requests for materials** should be addressed to Francesco Salvi.

# Reporting Summary

## Statistics

For all statistical analyses, confirm that the following items are present in the figure legend, table legend, main text, or Methods section.

| n/a | Confirmed | |
|---|---|---|
| ☐ | ☒ | The exact sample size ($n$) for each experimental group/condition, given as a discrete number and unit of measurement |
| ☐ | ☒ | A statement on whether measurements were taken from distinct samples or whether the same sample was measured repeatedly |
| ☐ | ☒ | The statistical test(s) used AND whether they are one- or two-sided<br>*Only common tests should be described solely by name; describe more complex techniques in the Methods section.* |
| ☐ | ☒ | A description of all covariates tested |
| ☐ | ☒ | A description of any assumptions or corrections, such as tests of normality and adjustment for multiple comparisons |
| ☐ | ☒ | A full description of the statistical parameters including central tendency (e.g. means) or other basic estimates (e.g. regression coefficient) AND variation (e.g. standard deviation) or associated estimates of uncertainty (e.g. confidence intervals) |
| ☐ | ☒ | For null hypothesis testing, the test statistic (e.g. $F$, $t$, $r$) with confidence intervals, effect sizes, degrees of freedom and $P$ value noted<br>*Give P values as exact values whenever suitable.* |
| ☒ | ☐ | For Bayesian analysis, information on the choice of priors and Markov chain Monte Carlo settings |
| ☒ | ☐ | For hierarchical and complex designs, identification of the appropriate level for tests and full reporting of outcomes |
| ☐ | ☒ | Estimates of effect sizes (e.g. Cohen's $d$, Pearson's $r$), indicating how they were calculated |

*Our web collection on statistics for biologists contains articles on many of the points above.*

## Software and code

Policy information about availability of computer code

| | |
|---|---|
| Data collection | Data were collected via a custom web-based application based on the Empirica v1.9.5. framework. Code can be found at https://github.com/epfl-dlab/debategpt. |
| Data analysis | Data analysis was performed using Python version 3.11 and R version 4.3.1. We also used LIWC-22 to extract textual features. Code can be found at https://github.com/epfl-dlab/debategpt. |

For manuscripts utilizing custom algorithms or software that are central to the research but not yet described in published literature, software must be made available to editors and reviewers. We strongly encourage code deposition in a community repository (e.g. GitHub). See the Nature Portfolio guidelines for submitting code & software for further information.

## Data

Policy information about availability of data

All manuscripts must include a data availability statement. This statement should provide the following information, where applicable:

- Accession codes, unique identifiers, or web links for publicly available datasets
- A description of any restrictions on data availability
- For clinical datasets or third party data, please ensure that the statement adheres to our policy

The debate dataset collected for our study is publicly available at https://huggingface.co/datasets/frasalvi/debategpt.

# Research involving human participants, their data, or biological material

Policy information about studies with human participants or human data. See also policy information about sex, gender (identity/presentation), and sexual orientation and race, ethnicity and racism.

| | |
|---|---|
| Reporting on sex and gender | We collected information about gender in our initial demographic survey, asking participants to self-report their gender identity. The final sample was composed of 49.6% Male, 47.7% Female, 2.5% Non binary/Non conforming, 0.1% Other. We included gender as a control in Supplementary Information Section 7, to exlude potential backdoors through demographics due to a randomly unbalanced assignment of participants to conditions. No gender-related effect was found. |
| Reporting on race, ethnicity, or other socially relevant groupings | We collected information about ethnicity in our initial demographic survey. The final sample was composed of 64.9% White/Caucasian, 16.7% Black or African American, 14.3% Asian or Pacific Islander, 9.7% Hispanic or Latinx, 0.8% Native American or American Indian, 0.9% Other. The categories were curated by researchers to roughly match the ones surveyed by the US Census Bureau. Notice that the percentages do not add up to 100%, as multiple answers were allowed. We included ethnicity as a control in Supplementary Information Section 7, to exlude potential backdoors through demographics due to a randomly unbalanced assignment of participants to conditions. No ethnicity-related effect was found. |
| Population characteristics | See "Research sample" below. |
| Recruitment | We recruited participants through Prolfic, allowing each worker to only participate in one debate to prevent skill disparity. To our knowledge, there were no significant sources of self-selection bias that would affect the results of our study. |
| Ethics oversight | The study protocol was approved by EPFL's Human Research Ethics Committee. |

Note that full information on the approval of the study protocol must also be provided in the manuscript.

# Field-specific reporting

Please select the one below that is the best fit for your research. If you are not sure, read the appropriate sections before making your selection.

☐ Life sciences  ☒ Behavioural & social sciences  ☐ Ecological, evolutionary & environmental sciences

For a reference copy of the document with all sections, see nature.com/documents/nr-reporting-summary-flat.pdf

# Behavioural & social sciences study design

All studies must disclose on these points even when the disclosure is negative.

| | |
|---|---|
| Study description | The study is a randomized controlled experiment where participants were matched either with another human participant or an LLM, carrying a written debate on a pre-assigned proposition in a between-subjects design. We recorded and analyzed quantitative measures of participants' agreement with their proposition before and after the debates. |
| Research sample | The research sample (N=900) was composed of Prolific users at least 18 years old and residing in the United States. The location requirement was motivated by the fact that most of our debate topics are deeply rooted in U.S. national issues, and would not resonate with different populations. The sample was selected at random, and hence is not necessarily representative of the U.S. population as a whole. Our final sample was 49.6% male, 47.7% female, 2.7% other, with the following age distribution: 11.3% 18-24, 34.1% 25-34, 23.7% 35-44, 17.3% 45-54, 8.7% 55-64, 4.8% 65+. |
| Sampling strategy | The sample was selected at random, using Prolific's standard sample algorithm. The sample size (150 debates per condition) was decided based on a separate small pilot and available resources, and was pre-registered before data collection. |
| Data collection | The experiment was conducted on a custom online platform, where participants completed the study on their own computers without researcher supervision. The researchers performing data collection and analysis were not blind to the conditions of the experiment. |
| Timing | Topic annotations were performed between 11 November and 22 November 2023. The debates were collected between December 2023 and April 2024. |
| Data exclusions | Coherently with our pre-registration, we excluded 20 debates where at least one human participant showed clear evidence of LLM usage or plagiarism, which explicitly contradicted our study instructions. Additionally, we excluded 13 debates were at least one participant provided unacceptable (empty texts, nonsensical or few-words arguments) or incomplete answers. The number of people involved in rejected debates is not counted towards the total number of participants reported in our manuscript (N=900), as the affected tasks were re-published on Prolific and completed by other workers. |
| Non-participation | A total of 377 participants recruited on Prolific returned the study or timed-out, effectively dropping out before completing it. Debates with dropped-out participants were consequently removed. As above, the number of affected participants does not count towards the total number reported in our manuscript (N=900). |
| Randomization | Participants were assigned at random into experimental groups. |

# Reporting for specific materials, systems and methods

We require information from authors about some types of materials, experimental systems and methods used in many studies. Here, indicate whether each material, system or method listed is relevant to your study. If you are not sure if a list item applies to your research, read the appropriate section before selecting a response.

## Materials & experimental systems

| n/a | Involved in the study |
|-----|----------------------|
| ☒ | ☐ Antibodies |
| ☒ | ☐ Eukaryotic cell lines |
| ☒ | ☐ Palaeontology and archaeology |
| ☒ | ☐ Animals and other organisms |
| ☒ | ☐ Clinical data |
| ☒ | ☐ Dual use research of concern |
| ☒ | ☐ Plants |

## Methods

| n/a | Involved in the study |
|-----|----------------------|
| ☒ | ☐ ChIP-seq |
| ☒ | ☐ Flow cytometry |
| ☒ | ☐ MRI-based neuroimaging |

## Plants

Seed stocks

*Report on the source of all seed stocks or other plant material used. If applicable, state the seed stock centre and catalogue number. If plant specimens were collected from the field, describe the collection location, date and sampling procedures.*

Novel plant genotypes

*Describe the methods by which all novel plant genotypes were produced. This includes those generated by transgenic approaches, gene editing, chemical/radiation-based mutagenesis and hybridization. For transgenic lines, describe the transformation method, the number of independent lines analyzed and the generation upon which experiments were performed. For gene-edited lines, describe the editor used, the endogenous sequence targeted for editing, the targeting guide RNA sequence (if applicable) and how the editor was applied.*

Authentication

*Describe any authentication procedures for each seed stock used or novel genotype generated. Describe any experiments used to assess the effect of a mutation and, where applicable, how potential secondary effects (e.g. second site T-DNA insertions, mosiacism, off-target gene editing) were examined.*

