## [Peer Review File · Nature Human Behaviour]

On the Conversational Persuasiveness of GPT-4

Corresponding Author: Mr Francesco Salvi

Version 0:

Decision Letter:

8th July 2024

Dear Mr. Salvi,

Thank you once again for your manuscript, entitled "On the Conversational Persuasiveness of Large Language Models: A Randomized Controlled Trial," and for your patience during the peer review process.

Your manuscript has now been evaluated by 3 reviewers, whose comments are included at the end of this letter. Although the reviewers find your work to be of interest, they also raise some important concerns. We are interested in the possibility of publishing your study in Nature Human Behaviour, but would like to consider your response to these concerns in the form of a revised manuscript before we make a decision on publication.

To guide the scope of the revisions, the editors discuss the referee reports in detail within the team, including with the chief editor, with a view to (1) identifying key priorities that should be addressed in revision and (2) overruling referee requests that are deemed beyond the scope of the current study. We hope that you will find the prioritised set of referee points to be useful when revising your study. Please do not hesitate to get in touch if you would like to discuss these issues further.

In particular, please ensure that you fully address the following (as well as all other reviewer comments):

- 1) Our preregistration policy (<https://www.nature.com/nathumbehav/editorial-policies/preregistration-policy>) requires that all preregistered analyses be reported exactly as preregistered, except if they are unfeasible or flawed. We ask that in revision you report the 3x3 design analyses, as preregistered. The 2x2 analyses can be retained. However, any differences between the two sets of analyses must be clearly discussed.
- 2) Address reviewer concerns regarding the framing and interpretation of your study, ensuring that claims match the design used and that limitations around ecological validity are thoroughly discussed.
- 3) Address concerns raised by Reviewers 1 and 2 regarding the results for participants assigned to promote positions they disagreed with. Although we do not expect new data collection, please ensure that the points the reviewers raised are discussed and addressed to the extent possible using existing data.
- 4) Ensure that statistically insignificant results are not interpreted as supportive of the hypotheses.

In sum, we invite you to revise your manuscript taking into account all reviewer and editor comments. We are committed to providing a fair and constructive peer-review process. Do not hesitate to contact us if there are specific requests from the reviewers that you believe are technically impossible or unlikely to yield a meaningful outcome.

We hope to receive your revised manuscript within two months. I would be grateful if you could contact us as soon as possible if you foresee difficulties with meeting this target resubmission date.

- Include a "Response to the editors and reviewers" document detailing, point-by-point, how you addressed each editor and

referee comment. If no action was taken to address a point, you must provide a compelling argument. When formatting this document, please respond to each reviewer comment individually, including the full text of the reviewer comment verbatim followed by your response to the individual point. This response will be used by the editors to evaluate your revision and sent back to the reviewers along with the revised manuscript.

- Highlight all changes made to your manuscript or provide us with a version that tracks changes.

Link Redacted

We look forward to seeing the revised manuscript and thank you for the opportunity to review your work. Please do not hesitate to contact me if you have any questions or would like to discuss these revisions further.

Sincerely,

██████████

████████████████████

██████████

Nature Human Behaviour

Reviewer expertise:

Reviewer #1: psychology, persuasion

Reviewer #2: persuasion, large language models

Reviewer #3: psychology, persuasion

REVIEWER COMMENTS:

Reviewer #1:

Remarks to the Author:

In this paper, "On the Conversational Persuasiveness of Large Language Models: A Randomized Controlled Trial," the authors examine how well ChatGPT 4 can persuade people's opinions on sociopolitical topics in a structured debate relative to a human. Additionally, they test whether having information that can serve to help personalize the arguments affects the messages' persuasiveness, ultimately finding that ChatGPT + personalized information is the most effective method for changing people's attitudes. Overall, I found the research to be very well written and conducted. Indeed, the authors wrote a clear paper that naturally flows from start to finish. They employed (created) an interesting methodology for testing their predictions. And they showed the effect across a range of different sociopolitical topics (from those that are more vs. less firmly held). They also did a good job citing the relevant literatures, which is not always the case in the fast-moving world on LLM research. With all of that said, I have two primary reactions that I think are worth considering.

First, and addressed by the authors themselves, is the artificiality of the debate paradigm. Admittedly, this is a problem for much of this initial LLM work as researchers are largely looking to test conceptual possibilities rather than ecologically realistic implementations. However, almost nowhere in the world do people engage in this sort of debate structure. Thus, although the authors do indeed show that tailored LLM messages are effective at shifting people's attitudes in this context, it necessarily needs to be better qualified to this specific claim. That is, I think the authors could more clearly emphasize in their introduction and discussion that this is more of a "proof of concept" study rather than one that shows LLM can change people's minds in natural online conversations, because the online conversations that generally happen (e.g., on Facebook, Twitter, etc.) are meaningfully different than this. Alternatively, maybe they could position this work as a potential intervention for helping to bridge political polarization on college campuses, in specific workshops, etc., but in short, the very unique design of this study limits the ecological validity of the findings. Building off this point, another threat to ecological validity is what's going on for participants who were asked to generate arguments for a position they don't agree with a priori. Although the authors claimed that this was needed for random assignment (which is not necessarily true as one could always assign participants to argue for their position and then simply test whether what side of the topic their position takes moderates the effect), incorporating the findings from these participants further limits the conclusions that can be drawn. That is, in this condition, is it really the LLM that is persuasive? Or is it seeing someone argue for their own side that matters? I believe the

authors test for moderation of this factor at the end of p. 10 and moving onto p. 11. However, the test they conducted wasn't entirely clear to me, whereas I would have simply liked to have seen their results either restricted to participants who generated arguments for a position they endorsed a priori and/or test prior endorsement of the stance they were asked to argue for as an additional factor in the analysis.

Second, it's unclear how much of their effect is due to people recognizing that their debate partner was an AI versus how much it was the arguments the AI generated. For example, the authors themselves comment on how it might be easier to "lose" to an AI debate partner. Thus, are people more readily willing to change their minds simply as a function of who they're speaking with or is it because AI is generating better arguments? Could the authors test whether the degree of logicity in the messages (from their LIWC coding) mediates attitude change? Although this wouldn't test for the personalization benefit, it could at least help to provide some empirics on why LLMs were generally more persuasive than humans. Relatedly, could the authors test whether participants' accuracy in guessing who the source is moderates the effect? For example, if people think they're speaking to an AI, does that lead to larger effects than if they think they're speaking to a human, irrespective of who their actual debate partner was? Another possibility would be to take an approach used by Costello, Pennycook, and Rand (2024) and recontact participants to see if their attitudes maintained the observed change over a greater degree of time. If this was true for the GPT + personalized condition (or just the GPT conditions overall) this might suggest that it really was something about the quality of the arguments rather than the perceived source of the arguments that matter for the effects.

Overall, again, I think the research is very well written and conducted; however, ultimately, it's unclear to me what the degree of contribution this study offers beyond previous research. That is, although it uses an interesting and unique debate design similar to Costello et al., the idea that LLMs can generate effective, personalized content has already been shown by Matz et al. (2024) and Simchon et al. (2024). Although the recent Hackenburg and Margetts (2024) paper shows the opposite takeaway, there are a number of concerns with their methodology that help to explain the null results. Thus, I think what this paper needs to do to make an advance for the literature is something along the following: (a) help explain why personalized LLM messages are more effective than personalized human messages, (b) help explain why Hackenburg and Margetts found non-significant personalization results when using similar demographic information for their LLMs as was employed in this research, (c) show how this paradigm can be used in a real-world setting to reduce political polarization, improve dialogues, etc., or (d) provide more granular insight on the persuasive influence of knowing that something is generated by AI versus the persuasive influence of the content AI generates itself.

I thank the authors for the opportunity to read this paper and wish them the best of luck in this work!

Reviewer #2:

Remarks to the Author:

General comment:

The study is based on a very innovative study design, it is interesting to read, and shows that AI is more persuasive than humans. I found this manuscript interesting, and I suggest the editors to publish it, provided the authors consider the minor points raised below. It is worth mentioning that a very exciting side-aspect of the study is that personalized AI arguments with humans are significantly more persuasive than non-personalized ones, albeit the authors find no trace of a "signature" that could explain why. This tells us, once more, how scary and yet fascinating is AI as a persuasive communicator.

Minor comments:

1. From line 082 to line 085 the authors mention that "Current work has explored the potential of AI-powered persuasion by comparing texts authored by humans and LLMs, finding that modern language models can generate content perceived as at least on par and often more persuasive than human-written message". Citation n. 42 (Spitale et al. Science Advances) can also be added here, as the main focus of that paper is the ability of GPT-3 to create disinformation that is more compelling than human-generated disinformation.
2. It is not clear what the authors mean by "every 5 minutes" when they explain that after completing a demographic survey, participants are assigned to one treatment condition. This should be better explained in the results section and in Fig.1 legend.
3. In lines 271-272, authors say that "On the other side, humans use more first-person singular and second-person pronouns and produce longer but easier-to-read texts". It is not sure what they mean by "easier-to-read" and how they measure this. Arguably, a more compelling text is one that is easier to understand, thus in this case easier to read. Adjust the discussion accordingly in lines 436-440.
4. It would be useful to expand the discussion around lines 273-307. The authors touch upon the relevance of prompt engineering in the study design. Arguably, all conclusions in the study depend on the specific prompt used. For example, authors underline that GPT-4 is using a language based on logic reasoning and knowledge, whereas humans display more appeals to support and trust. This can be changed if GPT-4 is instructed to appeal to support and trust, thus potentially further enhancing its ability to persuade humans.
5. In line 460-462, the authors rightly mention that a limitation of the study is that human participants may not necessarily

believe the standpoint they advocate for. I would make a stronger argument and stress more this limitation of the study, especially as a way to improve study design for a future study, that is to allow human-human interaction and engagement across those that have different starting opinions. These could have been done by design by matching those that, eg, strongly agreed vs strongly disagreed about a given topic. Despite the authors find the effect of opponents' agreement is non-significant, with the sample size available is likely hard to determine whether the analysis is sufficiently robust, and whether there are topic-based differences. The limited conversational structure, additionally, may not give sufficient time to develop nuanced debates starting from very different positions.

6. As an additional limitation, it might be relevant to mention that anonymized textual interactions are not the same as non-anonymized textual interactions between humans, and in person interactions between humans. This is thus just a step forward in understanding the complexities of AI-driven persuasiveness versus human-human interactions.

7. In lines 493-494, the authors say that "we manually reviewed each debate and discarded all the instances where we detected clear evidence of LLM usage or plagiarism". How was this done, exactly? It should be mentioned, as previous studies and also the authors discussion in this manuscript highlights the difficulty in distinguishing between AI and human-written content.

8. The chosen human participants (through mTurk) could definitely affect the results, and this should be mentioned as a limitation of the study. For example, choosing the participants via eg. Facebook could have changed the results, or choosing not to pay/incentivate respondents. Or as an additional examples, choosing skilled communicators. The comparison between humans and AI cannot be representative in this study. Of course, it is impossible to get a representative sample of participants, that represents a spectrum of persuasive skills that mimics the spectrum of skills in the US. Nonetheless the participants background, intentions, and motivations, can determine whether humans are more or less persuasive than AI.

Reviewer #3:

Remarks to the Author:

The manuscript "On the Conversational Persuasiveness of Large

Language Models: A Randomized Controlled Trial" uses an experimental design to show how LLMs with access to basic sociodemographic information can craft persuasive messages in natural debates that are more effective at changing their counterparts' opinions than human agents. They also explore how the content of LLMs differs from that of human agents, and how the perception of whether the opponent is an AI or a human is related to the effectiveness of persuasive arguments. I applaud the authors on an interesting and timely paper that is well designed, executed and written. I only have a few small comments and suggestions that I hope will help them further improve on an already excellent manuscript.

1) The opening sentence of the abstract reads "Can large language models (LLMs) create tailor-made, convincing arguments to promote false or misleading narratives online?". While I agree that this is one potential application of the authors' findings, I don't think it is what they test as part of their experimental setup. As the authors themselves acknowledge when stating that their study was conducted in a "controlled, harmless setting", the debated participants engaged in were not focused on crafting false or misleading information. I would instead simply refer LLMs ability to create tailor-made persuasive content.

2) I tripped over this sentence here a couple of times: "We find that 64.4% of the time, personalized LLM debaters were more persuasive than humans, given that they were not equally persuasive (81.2% relative increase in the odds of higher post-debate agreement; $p < 0.01$; $N = 900$).". Given that this is a key finding of the study, I would suggest trying to help the reader interpret it more easily. Maybe something along the lines of: "In those cases in which a direct comparison between a human and an AI-based debate resulted in significantly different levels of persuasion, the AI won against the human in 64.4% of cases." (that's at least what I understood from the SI explanation which was very helpful!). But this also made me wonder how often the AI wins overall (including the comparisons with no significant differences)? This would be interesting for policy makers.

3) There are a few times in the manuscript when significance is used rather liberally (e.g. "Despite that, the effect of Human-AI, personalized debates remains strong even for the High-Strength cluster (+64.2%, [-14.7%, +216.1%], $p = 0.14$), albeit with a lower statistical significance."; this is not really lower statistical significance but not statistically significant). I would recommend updating this throughout to only refer to effects as significant when they meet the $p < .05$ threshold.

4) The authors convinced me that their choice of Partial Proportional Odds Models is the way to go. However, I believe that much fewer readers will be familiar with these models (including myself). If the effects are robust, I would expect them to replicate with a regular linear regression analysis (which seems reasonable as the difference scores should be somewhat more normally distributed, I assume?). If the effects remain the same, it seems worth adding a footnote to say so.

5) The SI included all the relevant information on how the LLMs were prompted, but I would recommend pulling some of that information into the main manuscript. For example, I think it is critical to tell readers how the personalization of the LLM worked. As the authors note, there are contradictory findings reported in the literature, and one big factor determining the effectiveness of personalization is presumably the specific prompting strategies deployed across studies. Having the information in the main manuscript will simply make it a lot easier for readers to evaluate and compare.

6) I believe it would be worth adding to the limitations section of the discussion that the human debaters were all lay people

and that future research should investigate whether the same effects would be observed for experts (e.g. people working in communications or for political campaigns). This is both true for the main finding, but also for some of the other intriguing findings, such as the fact that personalization in highly opinionated topics backfires for humans, or that messages are always more persuasive when the receiver believes the opponent to be an AI.

7) How did the authors choose the specific LIWC categories to investigate? To my knowledge there are over 177 dimensions in the latest LIWC version, and it is unclear whether the authors had a priori expectations or post hoc focused on the dimensions with the largest differences. Both approaches are perfectly reasonable, but it would be helpful to know.

8) A few smaller things (mostly personal preferences, so please feel free to ignore):

a. I would stay clear of value judgements such as “harmless” when describing the research setting. I don’t think it’s necessary.

b. Sometimes the language might be a little bit too casual (e.g. “These obstacles might soon crumble”)

c. I didn’t understand this sentence “The efficacy of microtargeting has been questioned because it relies on effect heterogeneity, i.e., that specific groups of people respond differently to the same inputs [13–15].” Why is the reliance on heterogeneity a problem? This is clearly my ignorance, but it seems to me that heterogeneity is the feature that enables personalization in the first place?

d. The current manuscripts states: “opponents have access to anonymized information about participants” I would replace anonymized information with sociodemographic information which is both more descriptive and more accurate I believe.

e. Figure 4. Are those relative frequencies?

f. The color scheme confused me a little bit, especially as only one color had a legend. I don’t think the colors are needed at all, and I wonder if it would be helpful to merge the two figures to have both frequencies and effects in the same graph (maybe with frequencies in brackets)

Thank you again for letting me read this intriguing manuscript!

Version 1:

Decision Letter:

Our ref: NATHUMBEHAV-24051824A

20th December 2024

Dear Dr. Salvi,

Thank you for submitting your revised manuscript “On the Conversational Persuasiveness of Large Language Models: A Randomized Controlled Trial” (NATHUMBEHAV-24051824A). I apologize for the long delay in sending this decision on your revised manuscript.

Your study has now been seen by the original referees, and their comments are below. As you can see, the reviewers find that the paper has improved in revision. We will therefore be happy in principle to publish it in Nature Human Behaviour, pending minor revisions to satisfy the referees’ final requests and to comply with our editorial and formatting guidelines.

We are now performing detailed checks on your paper and will send you a checklist detailing our editorial and formatting requirements within approximately three weeks. (Please note that there may be a slight delay due to the upcoming holidays.) Please do not upload the final materials and make any revisions until you receive this additional information from us.

Sincerely,

██████████

████████████████████

████████████████████

Nature Human Behaviour

Reviewer #1 (Remarks to the Author):

In this second-round review of “On the Conversational Persuasiveness of Large Language Models: A Randomized Controlled Trial,” the authors provided new citations, analyses, and conclusions in support of their findings that gen-AI personalized persuasion can outperform human persuasion in a debate context. Overall, this review will be short, because I thought they adequately tested the additional effects and mechanisms I suggested, and they provided the appropriate language to address some of the limitations of the current work. Like in the first round, I found this to be a very impressive paper, and I think it will make a noteworthy contribution to the literature. I only have a few last final suggestions.

It would be nice to have a couple examples of what sociodemographic information was provided to the personalized

conditions. It doesn't need to be exhaustive but given the ongoing debate about what kinds of variables are most effective in generating personalized persuasion, it would be nice to have a parenthetical or even a sentence providing a little more context on that. For example, I had originally thought the sociodemographic information was simply age, gender, and race; however, in SI-5, I learned that it included political identification. In short, I think it would help to advance the broader discussion on personalized persuasion if the authors mentioned in the primary text a couple examples of that sociodemographic information, how many pieces of information there was in total, and whether the humans/AI had access to all of that information or simply a random subset of it. In this vein, it might be worth mentioning in the General Discussion how the effects for AI-personalized might have been even stronger if it had been provided with even more meaningful psychological variables (e.g., opponents' moral bases).

I really appreciated the discussion in the General Discussion about the study being a "proof of concept" with real world corollaries. I thought a single sentence to this effect in the introduction could be valuable, too (e.g., "Although this study uses a carefully controlled debate paradigm, it nonetheless serves as a meaningful proof of concept for the form in which debates like these take place online, such as through synchronous discussions in comment threads on websites like Facebook and Reddit."). In this vein, I thought it could be important to mention in the General Discussion that the structure of the experiment might have put a premium on information that was "fact-based and logical," as this is how formal debates are commonly determined to have winners and losers. In contrast, when "debating" casually with a friend, research has found that using I-statements and personal narratives can be more effective. That is to say that the AI advantage might have been facilitated by logic-favoring context of the experimental design. At the same time, that is not to say that if an AI had been prompted to use personal narratives, it couldn't have outperformed humans in a more casual form of political discourse, too.

Again, overall, I found this paper to be very well written and comprehensive and think it will make an excellent contribution to both the work on gen-AI as well as personalized persuasion!

Reviewer #2 (Remarks to the Author):

I thank the authors for successfully addressing all the points I raised in the round of review. Great job and precise and commendable review process.

Reviewer #3 (Remarks to the Author):

Thank you very much for addressing all my suggestions and adding additional explanations both in the cover letter and the main manuscript.

Your response to my comment #5 (pull the personalization prompt to the main manuscript) is very helpful. It made me think that you could even play this up a little in the discussion -- only if you want, not necessary at all. What I mean is that the prompt is incredibly simple, which means that the effects you observe are based on minimal prompting (and prompt engineering presumably). Might be worth highlighting.

I have nothing else to add, congratulations on a great paper!

Responses to Reviewers' Comments for Manuscript

NATHUMBEHAV-24051824

**On the Conversational Persuasiveness of
Large Language Models: A Randomized
Controlled Trial**

Addressed Comments for Publication to

Nature Human Behaviour

by

Francesco Salvi, Manoel Horta Ribeiro, Riccardo Gallotti, Robert West

Authors' Response to Reviewer 1

General Comments. In this paper, “On the Conversational Persuasiveness of Large Language Models: A Randomized Controlled Trial”, the authors examine how well ChatGPT 4 can persuade people’s opinions on sociopolitical topics in a structured debate relative to a human. Additionally, they test whether having information that can serve to help personalize the arguments affects the messages’ persuasiveness, ultimately finding that ChatGPT + personalized information is the most effective method for changing people’s attitudes. Overall, I found the research to be very well written and conducted. Indeed, the authors wrote a clear paper that naturally flows from start to finish. They employed (created) an interesting methodology for testing their predictions. And they showed the effect across a range of different sociopolitical topics (from those that are more vs. less firmly held). They also did a good job citing the relevant literatures, which is not always the case in the fast-moving world on LLM research.

Response: We thank the reviewer for their kind words.

Comment 1

First, and addressed by the authors themselves, is the artificiality of the debate paradigm. Admittedly, this is a problem for much of this initial LLM work as researchers are largely looking to test conceptual possibilities rather than ecologically realistic implementations. However, almost nowhere in the world do people engage in this sort of debate structure. Thus, although the authors do indeed show that tailored LLM messages are effective at shifting people's attitudes in this context, it necessarily needs to be better qualified to this specific claim. That is, I think the authors could more clearly emphasize in their introduction and discussion that this is more of a "proof of concept" study rather than one that shows LLM can change people's minds in natural online conversations, because the online conversations that generally happen (e.g., on Facebook, Twitter, etc.) are meaningfully different than this. Alternatively, maybe they could position this work as a potential intervention for helping to bridge political polarization on college campuses, in specific workshops, etc., but in short, the very unique design of this study limits the ecological validity of the findings.

Response: We agree with the reviewer's point that our study is a proof of concept about LLMs' persuasive capabilities. Nonetheless, we argue that our debate paradigm can still capture the essence of many online interactions: in modern social networks, people often engage with each other almost synchronously, reacting to others' comments in real-time in a back-and-forth discussion (e.g., comments on Facebook, Reddit, or Instagram).

In that context, following the reviewer's suggestion, we now write in the discussion: *Second, our experimental design forces conversations to have a predetermined structure, strictly following the stages and rules of a debate. While we believe that our setup can still capture the essence of many online interactions, where people reply to each other in an almost synchronous fashion or react to others' comments in real time, it still targets an artificial environment that can significantly diverge from the dynamics of online conversations, which evolve spontaneously and unpredictably. Additionally, on our*

platform, conversations were entirely anonymized, making them significantly different from the normal conditions under which humans interact. Therefore, we acknowledge that the ecological validity of our findings is limited, as it is unclear how our results would generalize to natural discussions on social networks and other open online platforms. For that reason, our work should be seen as providing a proof of concept about LLMs' persuasive capabilities rather than a realistic evaluation of their persuasiveness in the wild, which remains an open question for future research.

Comment 2

Building off this point, another threat to ecological validity is what's going on for participants who were asked to generate arguments for a position they don't agree with a priori. Although the authors claimed that this was needed for random assignment (which is not necessarily true as one could always assign participants to argue for their position and then simply test whether what side of the topic their position takes moderates the effect), incorporating the findings from these participants further limits the conclusions that can be drawn. That is, in this condition, is it really the LLM that is persuasive? Or is it seeing someone argue for their own side that matters? I believe the authors test for moderation of this factor at the end of p. 10 and moving onto p. 11. However, the test they conducted wasn't entirely clear to me, whereas I would have simply liked to have seen their results either restricted to participants who generated arguments for a position they endorsed a priori and/or test prior endorsement of the stance they were asked to argue for as an additional factor in the analysis.

Response:

The reviewer is correct in pointing out that randomizing the debate side (PRO or CON) was not strictly needed for causal identification. Instead, we designed our study in this way for practical reasons: matching participants in real-time presents a notable logistical

challenge, and it would have been considerably more difficult to conduct our experiment if we additionally had to enforce matches between people on opposing sides of each issue. We updated the manuscript to clarify this point.

To address the reviewers’ concerns, we significantly extended the analysis previously described on pages 10/11, adding a new Supplementary Information section (SI sec. 13, “Prior agreement and side randomization”). In short, as suggested by the reviewer, we (i) included the pre-treatment agreement of each debate opponent $\tilde{A}_{opponent}^{pre}$ as an additional control in our model, and we (ii) repeated our main analysis restricting the dataset to participants arguing for a side aligned with their prior opinions, essentially simulating a design without side randomization. For simplicity’s sake, we report the main results in Figure 1 and Figure 2. When controlling for $\tilde{A}_{opponent}^{pre}$, we find its effect to be statistically non-significant ($p = 0.22$), while the treatment effects remain virtually unchanged compared with those from the main analysis. Additionally, in our replication on participants aligned with their assigned side, we still find the *Human-AI, personalized* effect to be strong and statistically significant ($p = 0.03$), despite the lower statistical power given by the smaller data size. Overall, side randomization does not seem to have a significant impact on the findings of our study. Remarkably, this also seems to suggest that people can play the role assigned to them with great credibility, even when defending positions they do not spontaneously agree with. We updated our manuscript to incorporate these points in the main discussion, while also stressing more that future work would be needed to further generalize our results while enforcing matches between people on opposing sides of each issue.

Figure 1: Regression results for the partial proportional odds model, including as a control the pre-treatment agreements of debate opponents. The reference category corresponds to *Human-Human* debates. Error bars represent 95% confidence intervals.

Figure 2: Regression results for the partial proportional odds model, restricted to data-points where participants' prior agreements are aligned with the debate side to which they were assigned. The reference category corresponds to *Human-Human* debates. Error bars represent 95% confidence intervals.

Comment 3

Second, it's unclear how much of their effect is due to people recognizing that their debate partner was an AI versus how much it was the arguments the AI generated. For example, the authors themselves comment on how it might be easier to "lose" to an AI debate partner. Thus, are people more readily willing to change their minds simply as a function of who they're speaking with or is it because AI is generating better arguments? Could the authors test whether the degree of logicity in the messages (from their LIWC coding) mediates attitude change? Although this wouldn't test for the personalization benefit, it could at least help to provide some empirics on why LLMs were generally more persuasive than humans. Relatedly, could the authors test whether participants' accuracy in guessing who the source is moderates the effect? For example, if people think they're speaking to an AI, does that lead to larger effects than if they think they're speaking to a human, irrespective of who their actual debate partner was? Another possibility would be to take an approach used by Costello, Pennycook, and Rand (2024) and recontact participants to see if their attitudes maintained the observed change over a greater degree of time. If this was true for the GPT + personalized condition (or just the GPT conditions overall) this might suggest that it really was something about the quality of the arguments rather than the perceived source of the arguments that matter for the effects.

Response: We thank the reviewer for their insightful observation. We agree it would be interesting to clarify whether the effects can be explained simply by looking at who participants believed was their debate opponent. However, we stress that, with the data at hand, it is impossible to reach a conclusive answer to the question of how much those beliefs mediate the effects precisely because it is unclear if a causal relationship exists between participants' perceptions and shifts in agreements. Quoting from the manuscript, *"it is unclear whether the difference in agreement changes is motivated by participants' beliefs about their opponent or whether, conversely, those beliefs are caused*

Figure 3: Regression results for the partial proportional odds model, including *Perceived Opponent*. The reference category corresponds to *Human-Human* debates, where participants believed their opponent to be a Human. Error bars represent 95% confidence intervals.

by opinion changes. For example, participants could be more lenient in changing their agreement score towards their opponent when they believe they are facing an AI, because having a human on the other side makes it unconsciously easier to accept that they have somewhat lost the debate. Conversely, participants could also believe that their opponent was an AI because of how well their arguments were written.” In other words, we have no way of determining whether the variable *Perceived Opponent* is a mediator or a collider between the treatment and \tilde{A}^{post} , and controlling for it could thus disrupt the identification of causal effects (Cinelli et al., 2022). Nevertheless, we included in the manuscript (cf. Supplementary Information 12, “Perceived Opponent”) a regression that controls for *Perceived Opponent* to gain non-causal insights about its role, i.e., how much it correlates with the outcome. For convenience, we report the results of this regression in Figure 3. Citing from the Supplementary Information, *Perceived Opponent appears to play a substantial role, with a significant increase in agreement changes associated with participants that believe they were debating with an AI (odds of greater agreement with opponents +37.4%, [0.03%, 0.83%], $p = 0.03$)*. However, it is important to note

that the treatment effects remain similar to the ones resulting from our main analysis. Particularly, the *Human-AI, personalized* condition still has a strong and significant effect (+70.2%, [+17.8%, +146.0%], $p < 0.01$). Therefore, even if *Perceived Opponent* was a mediator (i.e., if the changing opinion was partly a function of who participants believed to be speaking with), it would not be sufficiently strong to explain the treatment effects. In other words, the direct effect of the treatment, which could arguably be interpreted as the capability of AI to generate better arguments, remains significant even when factoring out the indirect effect mediated by participants' perceptions. Building off this point, we tried to analyze other potential factors that could mediate the treatment effect, but we did not find any single mediator with a significant effect. When controlling for the degree of logicity, as the reviewer suggested, we find that the *Analytic* feature from LIWC increases the odds of higher post-treatment agreement by only +1.4% ($p = 0.97$) while leaving virtually unchanged all the treatment effects. Similar results were obtained with other individual features from LIWC and other textual metrics.

Unfortunately, our sample size does not give us enough statistical power to perform a proper mediation analysis that would simultaneously account for all those features. As we mentioned in our discussion, it thus remains an open question to establish why LLMs are more persuasive than humans. We updated the manuscript to include some of the points mentioned in this response and to provide a deeper interpretation of the regression in Supplementary Information 12. Finally, we agree with the reviewer that the approach of recontacting participants could provide additional evidence supporting LLMs' intrinsic persuasive capabilities, but we believe that it would be out of the scope of our work and better suited for a follow-up study. From a practical standpoint, it has been more than 8 months since we started collecting data (much more than the 2 months from Costello et al., 2024), and many of our original participants are no longer active on Prolific. Moreover, on a conceptual level, we argue that such an approach would address a different question than the one we tried to answer in our study, focusing more on the durability component of persuasive dialogues.

- Cinelli, C., Forney, A., & Pearl, J. (2022). A crash course in good and bad controls. *Sociological Methods & Research*, 53(3), 1071–1104. <https://doi.org/10.1177/00491241221099552>
- Costello, T. H., Pennycook, G., & Rand, D. G. (2024). Durably reducing conspiracy beliefs through dialogues with ai. <https://doi.org/10.31234/osf.io/xcwdn>

Comment 4

Overall, again, I think the research is very well written and conducted; however, ultimately, it's unclear to me what the degree of contribution this study offers beyond previous research. That is, although it uses an interesting and unique debate design similar to Costello et al., the idea that LLMs can generate effective, personalized content has already been shown by Matz et al. (2024) and Simchon et al. (2024). Although the recent Hackenburg and Margetts (2024) paper shows the opposite takeaway, there are a number of concerns with their methodology that help to explain the null results. Thus, I think what this paper needs to do to make an advance for the literature is something along the following: (a) help explain why personalized LLM messages are more effective than personalized human messages, (b) help explain why Hackenburg and Margetts found non-significant personalization results when using similar demographic information for their LLMs as was employed in this research, (c) show how this paradigm can be used in a real-world setting to reduce political polarization, improve dialogues, etc., or (d) provide more granular insight on the persuasive influence of knowing that something is generated by AI versus the persuasive influence of the content AI generates itself.

Response: To demonstrate the relevance of our contribution, we briefly comment on the previous works mentioned by the reviewer.

- In Matz et al., 2024, the authors study how personalized messages generated by

ChatGPT perform across different domains of persuasion and psychological profiles over the course of four studies. In Studies 1 and 2, AI-generated content is evaluated by asking participants to express their preference on a bipolar scale between messages tailored towards the opposite ends of one psychological characteristic. Study 3 exposes participants to single messages instead of pairs, but, similarly, compares ratings between messages customized to the high and low ends of one personality trait. Therefore, Studies 1-3 measure the effectiveness of accurate personalization vs. inaccurate personalization, using as a baseline a condition akin to the “False targeting” one from Hackenburg and Margetts, 2024. While the results interestingly showcase that LLMs can understand psychological profiles and successfully generate tailored messages, they arguably fall short of measuring the overall effectiveness of targeted messages since they never compare them with neutral ones. Additionally, all the messages used in Studies 1-3 are static: a single version of each text is generated for each dimension (high/low end of one psychological trait) before the experiments. Study 4 addresses those two points by dynamically generating unique messages for each individual and using a generic prompt as a comparison baseline. However, the design of Study 4 only considers two narrow applications of persuasion: ads to promote a weekend getaway to Rome or Nike sneakers. Out of those, the authors found a significant effect only for the Rome getaway ad. Therefore, despite being methodologically more robust, Study 4 also provides limited evidence about the persuasive capabilities of LLMs and does not generalize well to other settings.

- Simchon et al., 2024 focuses on political messages, studying the impact of micro-targeting with LLMs across two studies. In Study 1, the authors manually selected a set of 10 political ads from a larger pool of ads previously published on Facebook. They then used GPT to assign an “openness score” (one of the Big Five personality traits) to each ad, finding a significant persuasive effect when ads matched the Openness dimension of participants. In Study 2, they prompted LLMs to rephrase existing ads in their corpus to generate two variations, catering to people with

high or low openness scores. Again, they found a positive effect when ads matched participants' profiles, demonstrating that ChatGPT can be used as a supportive tool to automate the creation of tailored messages. However, both studies make little use of LLMs' generative capabilities: all the ads were based on pre-existing ones and were at most rephrased by ChatGPT with slight variations. It remains thus unclear how these results would generalize to messages entirely generated by LLMs, which is the scenario that raises most concerns amongst scholars in terms of societal implications (Bommasani et al., 2021). Additionally, similarly to Matz et al., 2024, the comparison was made between ads tailored to different openness scores, with the lack of a baseline condition that includes neutral, non-targeted messages. Again, we argue that this crucially limits the significance of the study findings, since it is not clear how much of the effect is simply due to the reduced persuasiveness of mismatched tailoring. Finally, the authors found significant but small effect sizes, showing only a marginal increase in personalized messages.

- Hackenburg and Margetts, 2024 integrated self-reported sociodemographic and political attributes into GPT-4 prompts, instructing it to persuade users on four political issues. Comparing messages with a control condition where participants' opinions were measured without exposure to any persuasive message, the authors found LLM-generated messages to be widely persuasive, but no significant differences between personalized and non-personalized messages, in contrast with the other studies mentioned above. A possible explanation for their null results is that interactions with LLMs were restricted to the exposure of a single message, a setup that significantly diminishes LLMs' potential capabilities compared to the conversational type of interactions for which modern language models are optimized.

In this context, our experiment provides for the first time broad evidence that, when provided with personal information, LLMs can be highly persuasive across a wide range of divisive political and social issues, showing remarkably strong effects even on highly polarized topics. On a methodological and conceptual level, our study introduces two

major innovations: (i) We deploy LLMs in a purely conversational setting, allowing for multi-turn discussions between human participants and LLMs. As we mentioned above, this kind of interaction is what modern language models are optimized for, and where they can express their full potential. While our debate setup is artificial and certainly has limitations in terms of ecological validity, we argue that it still provides the most realistic assessment of LLM persuasion to date, since people mostly interact with LLMs through open back-and-forth dialogues. (ii) We evaluate persuasive capabilities by comparing LLMs with real humans, showcasing how personalization is effective even against human participants with access to the same information. To the best of our knowledge, our study is the first randomized controlled trial that systematically compares LLMs and humans in live conversations.

Finally, we comment on the reviewer’s suggestions for improving our contributions:

- (a) Although we extensively investigated this point in our manuscript, we found no quantifiable differences between *Human-AI* and *Human-AI, personalized* debates across several textual and meta-textual dimensions (LIWC features, social dimensions, and usage of persuasive strategies). We speculated in our discussion about what could, thus, drive the difference in persuasive effects, hypothesizing that personalization is distinguished by the very choice of issues brought up during the debates. In Supplementary Information 5 (“Debate Examples”), we provided some examples to showcase how this mechanism could be unrolling. Ultimately, however, the question of why personalization brings such a strong persuasive improvement remains very open, needing further work to be properly answered.
- (b) As we argued above, we believe that a prominent explanation for the null results of Hackenburg and Margetts, 2024 is that in their study, interactions with LLMs were restricted to the exposure of a single message, significantly diminishing LLMs’ potential capabilities compared to conversational interactions.
- (c) We agree with the reviewer that implementing our paradigm in a real-world setting would be an extremely relevant addition, but we believe that designing an entirely new study aimed at reducing political polarization or improving online dialogues

would be outside the scope of this work. Nevertheless, we are interested in exploring those directions in future follow-up research.

- (d) We improved our discussion on this point (cf. Comment 3 for more details), clarifying that participants' perceptions of their debate opponent (i.e., whether they thought they were debating with a human or an AI) do not explain the persuasive effect of LLMs.

Bommasani, R., Hudson, D. A., Adeli, E., Altman, R., Arora, S., von Arx, S., Bernstein, M. S., Bohg, J., Bosselut, A., Brunskill, E., Brynjolfsson, E., Buch, S., Card, D., Castellon, R., Chatterji, N., Chen, A., Creel, K., Davis, J. Q., Demszky, D., . . . Liang, P. (2021). On the opportunities and risks of foundation models. <https://doi.org/10.48550/arXiv.2108.07258>

Hackenburg, K., & Margetts, H. (2024). Evaluating the persuasive influence of political microtargeting with large language models. *Proceedings of the National Academy of Sciences*, *121*(24), e2403116121. <https://doi.org/10.1073/pnas.2403116121>

Matz, S. C., Teeny, J. D., Vaid, S. S., Peters, H., Harari, G. M., & Cerf, M. (2024). The potential of generative ai for personalized persuasion at scale. *Scientific Reports*, *14*(1). <https://doi.org/10.1038/s41598-024-53755-0>

Authors' Response to Reviewer 2

General Comments. The study is based on a very innovative study design, it is interesting to read, and shows that AI is more persuasive than humans. I found this manuscript interesting, and I suggest the editors to publish it, provided the authors consider the minor points raised below. It is worth mentioning that a very exciting side-aspect of the study is that personalized AI arguments with humans are significantly more persuasive than non-personalized ones, albeit the authors find no trace of a “signature” that could explain why. This tells us, once more, how scary and yet fascinating is AI as a persuasive communicator.

Response: We thank the reviewer for their kind words and their endorsement.

Comment 1

1. From line 082 to line 085 the authors mention that “Current work has explored the potential of AI-powered persuasion by comparing texts authored by humans and LLMs, finding that modern language models can generate content perceived as at least on par and often more persuasive than human-written message”. Citation n. 42 (Spitale et al. Science Advances) can also be added here, as the main focus of that paper is the ability of GPT-3 to create disinformation that is more compelling than human-generated disinformation.

Response: We thank the reviewer for the suggestion. We added Spitale et al. to the citations for that claim.

Comment 2

2. It is not clear what the authors mean by “every 5 minutes” when they explain that after completing a demographic survey, participants are assigned to one treatment condition. This should be better explained in the results section and in Fig.1 legend.

Response: We slightly rephrased the legend of Fig.1, which now reads: “[...] (B) At intervals of 5 minutes, participants who have completed the survey are randomly assigned to one of four treatment conditions [...]”. In the main text, we specify that “At each clock trigger of a continuous 5-minute interval timer, all the participants that have completed the survey were randomly assigned to a treatment condition and thus matched with an appropriate opponent”. In other words, we mean that the matching is performed every 5 minutes, considering all the participants who have completed their survey by that time.

Comment 3

3. In lines 271-272, authors say that “On the other side, humans use more first-person singular and second-person pronouns and produce longer but easier-to-read texts”. It is not sure what they mean by “easier-to-read” and how they measure this. Arguably, a more compelling text is one that is easier to understand, thus in this case easier to read. Adjust the discussion accordingly in lines 436-440.

Response: We measured *easiness to read* using the Flesch Reading Ease score, a widely used metric that considers the average number of words per sentence and the average number of syllables per word (Flesch, 1948). We clarified that on lines 316-317, as well as in the legend of Fig.4, and we added in both a reference to Supplementary Information 10 (“Textual Analysis”), where we provide additional information about the meaning of each textual feature.

Flesch, R. (1948). A new readability yardstick. *Journal of Applied Psychology*, 32(3), 221–233. <https://doi.org/10.1037/h0057532>

Comment 4

4. It would be useful to expand the discussion around lines 273-307. The authors touch upon the relevance of prompt engineering in the study design. Arguably, all conclusions in the study depend on the specific prompt used. For example, authors underline that GPT-4 is using a language based on logic reasoning and knowledge, whereas humans display more appeals to support and trust. This can be changed if GPT-4 is instructed to appeal to support and trust, thus potentially further enhancing its ability to persuade humans.

Response: We agree with the reviewer about the importance of prompt engineering and the usefulness of emphasizing its role in the linguistic style exhibited by GPT-4. To avoid overloading the Results section, instead of expanding lines 273-307, we added the following paragraph in the Discussion (lines 483-487): *“Additionally, we emphasize how prompting plays a big role in the textual signature exhibited by LLMs. Future work could experiment with prompts that instruct GPT-4 to be less reliant on logical reasoning and showcase more appeals to support and trust, mimicking the style of human debaters and potentially enhancing its persuasive capabilities.”*

Comment 5

5. In line 460-462, the authors rightly mention that a limitation of the study is that human participants may not necessarily believe the standpoint they advocate for. I would make a stronger argument and stress more this limitation of the study, especially as a way to improve study design for a future study, that is to allow human-human interaction and engagement across those that have different starting opinions. These could have been done by design by matching those that, eg, strongly agreed vs strongly disagreed about a given topic. Despite the authors find the effect of opponents' agreement is non-significant, with the sample size available is likely hard to determine whether the analysis is sufficiently robust, and whether there are topic-based differences. The limited conversational structure, additionally, may not give sufficient time to develop nuanced debates starting from very different positions.

Response: To address the reviewer's concerns, we significantly extended the analysis previously described in lines 460-462, adding a new Supplementary Information section (SI sec. 13, "Prior agreement and side randomization"). There, we (i) included the pre-treatment agreement of each debate opponent $\tilde{A}_{opponent}^{pre}$ as an additional control in our model, and (ii) repeated our main analysis, restricting the dataset to participants arguing for a side aligned with their prior opinions, essentially simulating a design without side randomization. For a summary of the main results of these new analyses, we refer to our response to Reviewer #1, Comment 2. Overall, we show that side randomization does not significantly impact the findings of our study, also suggesting that people can play the role assigned to them with great credibility even when defending positions they do not spontaneously agree with. We updated our manuscript to incorporate these points in the main discussion. At the same time, as the reviewer suggested, we also stressed in our discussion that future work is needed to generalize our results further while enforcing matches between people on opposing sides of each issue, explaining in the Supplementary Information the logistical challenge this would entail.

Comment 6

6. As an additional limitation, it might be relevant to mention that anonymized textual interactions are not the same as non-anonymized textual interactions between humans, and in person interactions between humans. This is thus just a step forward in understanding the complexities of AI-driven persuasiveness versus human-human interactions.

Response: We agree that anonymity introduces a relevant difference from real human interactions that should be mentioned in our limitations. We rewrote part of our discussion to emphasize the limited ecological validity of our design, including also this point:

Second, our experimental design forces conversations to have a predetermined structure, strictly following the stages and rules of a debate. While we believe that our setup can still capture the essence of many online interactions, where people reply to each other in an almost synchronous fashion or react to others' comments in real time, it still targets an artificial environment that can significantly diverge from the dynamics of online conversations, which evolve spontaneously and unpredictably. Additionally, on our platform, conversations were entirely anonymized, making them significantly different from the normal conditions under which humans interact. Therefore, we acknowledge that the ecological validity of our findings is limited, as it is unclear how our results would generalize to natural discussions on social networks and other open online platforms. For that reason, our work should be seen as providing a proof of concept about LLMs' persuasive capabilities rather than a realistic evaluation of their persuasiveness in the wild, which remains an open question for future research.

Comment 7

7. In lines 493-494, the authors say that “we manually reviewed each debate and discarded all the instances where we detected clear evidence of LLM usage or plagiarism”. How was this done, exactly? It should be mentioned, as previous studies and also the authors discussion in this manuscript highlights the difficulty in distinguishing between AI and human-written content.

Response: We agree with the reviewer that this point requires additional clarification. We expanded the cited paragraph as follows:

Regardless, coherently with our pre-registration, we manually reviewed each debate and excluded 20 debates where at least one human participant showed clear indications of LLM usage (unrealistic WPM values, blatant evidence of ChatGPT’s standard writing style) and plagiarism (as detected by DupliChecker¹). Additionally, we excluded 13 debates where at least one participant provided unacceptable (empty texts, nonsensical or few-word arguments) or incomplete answers. The number of people involved in rejected debates is not counted towards the total number of participants (N=900), as the affected tasks were re-published on Prolific and completed by other workers.

We acknowledge that “blatant evidence of ChatGPT’s standard writing style” is still a rather generic description, which somewhat conflicts with our previous discussion about the difficulty in distinguishing between AI and human-written content. However, it has also been shown that ChatGPT with standard prompting exhibits in some cases a rather unique and distinctive writing style, with an excess of specific marker words and stylistic structures (Kobak et al., 2024). In our manual review, we only excluded instances where such elements were present with a frequency that can be associated with ChatGPT beyond a reasonable doubt. To illustrate that, we provide a concrete example from a debate that was excluded from the final dataset, reporting in the following the Opening argument of a participant who was assigned to the PRO side of the proposition “Should Governments Have the Right to Censor the Internet?”.

¹<https://www.duplichecker.com/>

[REDACTED]

Several participants excluded from the study, including the author of the argument reported above, later admitted through Prolific direct messaging to having used LLMs in their writing.

Kobak, D., González-Márquez, R., Horvát, E.-Á., & Lause, J. (2024). Delving into chatgpt usage in academic writing through excess vocabulary. <https://doi.org/10.48550/ARXIV.2406.07016>

Comment 8

8. The chosen human participants (through mTurk) could definitely affect the results, and this should be mentioned as a limitation of the study. For example, choosing the participants via eg. Facebook could have changed the results, or choosing not to pay/incentivate respondents. Or as an additional examples, choosing skilled communicators. The comparison between humans and AI cannot be representative in this study. Of course, it is impossible to get a representative sample of participants, that represents a spectrum of persuasive skills that mimics the spectrum of skills in the US. Nonetheless the participants background, intentions, and motivations, can determine whether humans are more or less persuasive than AI.

Response: We agree that the choice of recruiting participants through a crowd-working platform can significantly impact the results of our study, and should be mentioned in its limitations.

We expanded our discussion to include the following paragraph: *“Fourth, our experiment engaged human participants recruited through Prolific, who received financial incentives for completing debates and were aware of being in a controlled experimental environment.*

Although previous research has found Prolific to have the best data quality amongst competitors and research done using Prolific to be often generalizable (Douglas et al., 2023; Peer et al., 2021; Redmiles et al., 2019), the pool of workers active on the platform still differs significantly in their sociodemographic distribution from both the overall U.S. population and the userbase of other online platforms and social media. Therefore, future work is needed to understand whether our findings can be reproduced using a more representative sample that accurately mimics the overall spectrum of human persuasive skills. Additionally, it would be interesting to include human experts in our comparison, such as individuals involved in competitive debating, political campaigns, or public communication. This would be especially relevant to validate some of the secondary findings of our study, such as the backfire effect observed overall and particularly in the Human-Human, personalized condition for highly polarized topics, as well as the correlation between changes in agreement and participants' beliefs about their debate opponents.”

-
- Douglas, B. D., Ewell, P. J., & Brauer, M. (2023). Data quality in online human-subjects research: Comparisons between mturk, prolific, cloudresearch, qualtrics, and sona (J. S. Hallam, Ed.). *PLOS ONE*, *18*(3), e0279720. <https://doi.org/10.1371/journal.pone.0279720>
- Peer, E., Rothschild, D., Gordon, A., Evernden, Z., & Damer, E. (2021). Data quality of platforms and panels for online behavioral research. *Behavior Research Methods*, *54*(4), 1643–1662. <https://doi.org/10.3758/s13428-021-01694-3>
- Redmiles, E. M., Kross, S., & Mazurek, M. L. (2019). How well do my results generalize? comparing security and privacy survey results from mturk, web, and telephone samples. *2019 IEEE Symposium on Security and Privacy (SP)*, 1326–1343. <https://doi.org/10.1109/SP.2019.00014>

Authors' Response to Reviewer 3

General Comments. The manuscript “On the Conversational Persuasiveness of Large Language Models: A Randomized Controlled Trial” uses an experimental design to show how LLMs with access to basic sociodemographic information can craft persuasive messages in natural debates that are more effective at changing their counterparts’ opinions than human agents. They also explore how the content of LLMs differs from that of human agents, and how the perception of whether the opponent is an AI or a human is related to the effectiveness of persuasive arguments. I applaud the authors on an interesting and timely paper that is well designed, executed and written. I only have a few small comments and suggestions that I hope will help them further improve on an already excellent manuscript.

Response: We thank the reviewer for their kind words.

Comment 1

1) The opening sentence of the abstract reads “Can large language models (LLMs) create tailor-made, convincing arguments to promote false or misleading narratives online?”. While I agree that this is one potential application of the authors’ findings, I don’t think it is what they test as part of their experimental setup. As the authors themselves acknowledge when stating that their study was conducted in a “controlled, harmless setting”, the debated participants engaged in were not focused on crafting false or misleading information. I would instead simply refer LLMs ability to create tailor-made persuasive content.

Response: We agree with the reviewer that that sentence alludes more to a potential implication of our findings rather than to our specific setup. We rephrased it as: “*Can large language models (LLMs) craft tailor-made, convincing arguments to change people’s minds on polarizing political issues?*”.

Comment 2

2) I tripped over this sentence here a couple of times: “We find that 64.4% of the time, personalized LLM debaters were more persuasive than humans, given that they were not equally persuasive (81.2% relative increase in the odds of higher post-debate agreement; $p < 0.01$; $N = 900$).” Given that this is a key finding of the study, I would suggest trying to help the reader interpret it more easily. Maybe something along the lines of: “In those cases in which a direct comparison between a human and an AI-based debate resulted in significantly different levels of persuasion, the AI won against the human in 64.4% of cases.” (that’s at least what I understood from the SI explanation which was very helpful!). But this also made me wonder how often the AI wins overall (including the comparisons with no significant differences)? This would be interesting for policy makers.

Response: We apologize if the sentence was not clear. We tried to make it easier to understand by rephrasing it as follows: *“We find that, in debates where AI and human participants were not equally persuasive, LLMs with access to personalization won over humans 64.4% of the time.”*

Unfortunately, we do not have an easy way to systematically compute the probability of AI winning an overall comparison using our statistical estimates. Indeed, using the same notation as in Supplementary Information 3 (“Model specification”, lines 1541-1564), if we don’t assume that $\tilde{A}_i^{post} \neq \tilde{A}_j^{post}$, then the probabilities in Eq. (7) and (8) become:

$$P(\tilde{A}_j^{post} > \tilde{A}_i^{post}) = \sum_{a=1}^4 P(\tilde{A}_j^{post} > a, \tilde{A}_i^{post} \leq a | T_j = 1, T_i = 0) \quad (1)$$

$$P(\tilde{A}_j^{post} < \tilde{A}_i^{post}) = \sum_{a=1}^4 P(\tilde{A}_j^{post} \leq a, \tilde{A}_i^{post} > a | T_j = 1, T_i = 0) \quad (2)$$

Consequently, the odds κ that j was more persuaded than i would be given by:

$$\kappa = \frac{P(\tilde{A}_j^{post} > \tilde{A}_i^{post})}{P(\tilde{A}_j^{post} < \tilde{A}_i^{post})} = \frac{\sum_{a=1}^4 P(\tilde{A}_j^{post} > a | T = 1) P(\tilde{A}_i^{post} \leq a | T = 0)}{\sum_{a=1}^4 P(\tilde{A}_j^{post} \leq a | T = 1) P(\tilde{A}_i^{post} > a | T = 0)} \quad (3)$$

Contrarily to the case of $\tilde{A}_i^{post} \neq \tilde{A}_j^{post}$, the presence of summations makes it impossible to simplify that expression to a quantity that can be computed with our estimates.

Comment 3

3) There are a few times in the manuscript when significance is used rather liberally (e.g. “Despite that, the effect of Human-AI, personalized debates remains strong even for the High-Strength cluster (+64.2%, [-14.7%, +216.1%], $p = 0.14$), albeit with a lower statistical significance.”; this is not really lower statistical significance but not statistically significant). I would recommend updating this throughout to only refer to effects as significant when they meet the $p < .05$ threshold.

Response: We updated that sentence to read “*albeit in a non-significant fashion*” instead of “*albeit with a lower statistical significance*”. We re-examined our discussion of statistical significance throughout the manuscript, but we did not find other instances where effects are described as “marginally significant” or a formulation along similar lines for $p \geq 0.05$.

Comment 4

4) The authors convinced me that their choice of Partial Proportional Odds Models is the way to go. However, I believe that much fewer readers will be familiar with these models (including myself). If the effects are robust, I would expect them to replicate with a regular linear regression analysis (which seems reasonable as the difference scores should be somewhat more normally distributed, I assume?). If the effects remain the same, it seems worth adding a footnote to say so.

Response: We agree with the reviewer that it is likely that most readers will not be familiar with the Partial Proportional Odds Model. Unfortunately, it is tricky to directly compare the results of a linear regression, where the coefficients associated with each treatment express its average increase in the outcome, and the estimates of an ordinal model, which at most can be connected to the relative change in the odds of \tilde{A}^{post} assuming higher values. Nevertheless, we added a new Supplementary Information

section (SI sec. 7, “Linear Regression”) where we replicated our main analysis using simple linear regression, to provide some basic insights about the effectiveness of our treatments. For convenience, we report in Figure 4 the regression results. We find that, on average, *Human-AI, personalized* debates are associated with a 0.36 ([0.12, 0.60], $p < 0.01$) increase compared with the *Human-Human* condition in the difference $\tilde{A}^{post} - \tilde{A}^{pre}$. In contrast, the other two treatments have again non-significant coefficients, similarly to the results discussed in the main text.

Figure 4: Regression results for the linear regression $\tilde{A}^{post} - \tilde{A}^{pre} = \beta + \beta_{\mathbf{T}} \cdot \mathbf{T}$. We take as reference the *Human-Human*. Error bars represent 95% confidence intervals.

Comment 5

5) The SI included all the relevant information on how the LLMs were prompted, but I would recommend pulling some of that information into the main manuscript. For example, I think it is critical to tell readers how the personalization of the LLM worked. As the authors note, there are contradictory findings reported in the literature, and one big factor determining the effectiveness of personalization is presumably the specific prompting strategies deployed across studies. Having the information in the main manuscript will simply make it a lot easier for readers to evaluate and compare.

Response: We agree with the reviewer about the importance of prompting on the effectiveness of personalization. Given the relative simplicity of our prompting strategy, we added to our Procedure section (lines 180-183) the following sentence, reporting the key instruction provided to the LLMs for the personalized condition: *“When the LLM was provided with personal attributes, we instructed it to ‘astutely use this information to craft arguments that are more likely to persuade and convince your opponent of your stance’ (see Supplementary Information 2.5 for the complete prompts).”*

Comment 6

6) I believe it would be worth adding to the limitations section of the discussion that the human debaters were all lay people and that future research should investigate whether the same effects would be observed for experts (e.g. people working in communications or for political campaigns). This is both true for the main finding, but also for some of the other intriguing findings, such as the fact that personalization in highly opinionated topics backfires for humans, or that messages are always more persuasive when the receiver believes the opponent to be an AI.

Response: We agree that recruiting laypeople as human participants can significantly impact the results of our study and should be mentioned in its limitations.

We expanded our discussion to include the following paragraph: *“Fourth, our experiment engaged human participants recruited through Prolific, who received financial incentives for completing debates and were aware of being in a controlled experimental environment. Although previous research has found Prolific to have the best data quality amongst competitors and research done using Prolific to be often generalizable (Douglas et al., 2023; Peer et al., 2021; Redmiles et al., 2019), the pool of workers active on the platform still differs significantly in their sociodemographic distribution from both the overall U.S. population and the userbase of other online platforms and social media. Therefore, future work is needed to understand whether our findings can be reproduced using a more representative sample that accurately mimics the overall spectrum of human persuasive skills. Additionally, it would be interesting to include human experts in our comparison, such as individuals involved in competitive debating, political campaigns, or public communication. This would be especially relevant to validate some of the secondary findings of our study, such as the backfire effect observed overall and particularly in the Human-Human, personalized condition for highly polarized topics, as well as the correlation between changes in agreement and participants’ beliefs about their debate opponents.”*

Douglas, B. D., Ewell, P. J., & Brauer, M. (2023). Data quality in online human-subjects research: Comparisons between mturk, prolific, cloudresearch, qualtrics, and sona (J. S. Hallam, Ed.). *PLOS ONE*, 18(3), e0279720. <https://doi.org/10.1371/journal.pone.0279720>

Peer, E., Rothschild, D., Gordon, A., Evernden, Z., & Damer, E. (2021). Data quality of platforms and panels for online behavioral research. *Behavior Research Methods*, 54(4), 1643–1662. <https://doi.org/10.3758/s13428-021-01694-3>

Redmiles, E. M., Kross, S., & Mazurek, M. L. (2019). How well do my results generalize? comparing security and privacy survey results from mturk, web, and telephone

samples. *2019 IEEE Symposium on Security and Privacy (SP)*, 1326–1343. <https://doi.org/10.1109/SP.2019.00014>

Comment 7

7) How did the authors choose the specific LIWC categories to investigate? To my knowledge there are over 177 dimensions in the latest LIWC version, and it is unclear whether the authors had a priori expectations or post hoc focused on the dimensions with the largest differences. Both approaches are perfectly reasonable, but it would be helpful to know.

Response: We chose a subset of the corpus of LIWC features based on our a-priori expectations. In particular, as we added to the manuscript (in Supplementary Information 10, “Textual Analysis”), *“Our choice of features was informed by previous literature, including the ones that have been found to be most influential (Gligorić et al., 2021), as well as broad summary variables (Analytic, Clout, Authentic, Tone) that somewhat summarize different textual dimensions.”*

Gligorić, K., Lifchits, G., West, R., & Anderson, A. (2021). Linguistic effects on news headline success: Evidence from thousands of online field experiments (registered report protocol) (S. Lev-Ari, Ed.). *PLOS ONE*, *16*(9), e0257091. <https://doi.org/10.1371/journal.pone.0257091>

Comment 8

8) A few smaller things (mostly personal preferences, so please feel free to ignore):

- a. I would stay clear of value judgements such as “harmless” when describing the research setting. I don’t think it’s necessary.
- b. Sometimes the language might be a little bit too casual (e.g. “These obstacles might soon crumble”)
- c. I didn’t understand this sentence “The efficacy of microtargeting has been questioned because it relies on effect heterogeneity, i.e., that specific groups of people respond differently to the same inputs [13–15].” Why is the reliance on heterogeneity a problem? This is clearly my ignorance, but it seems to me that eterogeneity is the feature that enables ersonalization in the first place?
- d. The current manuscripts states: “opponents have access to anonymized information about participants” I would replace anonymized information with sociodemographic information which is both more descriptive and more accurate I believe.
- e. Figure 4. Are those relative frequencies?
- f. The color scheme confused me a little bit, especially as only one color had a legend. I don’t think the colors are needed at all, and I wonder if it would be helpful to merge the two figures to have both frequencies and effects in the same graph (maybe with frequencies in brackets).

Response: We thank the reviewer for their suggestions, which we tried to incorporate wherever possible. In particular,

- a. We tried to avoid value judgments, e.g., “novel”, “first”, “robust”, etc. However, in this case, we find that “harmless” is an important qualifier for our study, reflecting the fact that EPFL’s HREC (Human Research Ethics Committee) approved our

setup without highlighting any specific ethical risks. We believe it is important to emphasize this aspect of our setup since research on persuasion can easily lead to harmful outcomes, especially as it often includes deceiving aspects.

- b. We rephrased that sentence to: *“These limitations may soon fall off”*.
- c. The reviewer is correct in pointing out that the reliance on heterogeneity is not a problem per se, but rather the feature that enables personalization in the first place. What we meant is that, indeed, effect heterogeneity is an additional assumption that needs to be valid for personalization to work, something that, in general, is not given and has been debated in previous literature. We rephrased that sentence to: *“The efficacy of microtargeting has been questioned because it relies on the assumption of effect heterogeneity, i.e., that specific groups of people respond differently to the same inputs, a concept that has been disputed in previous literature.”*
- d. We replaced *“anonymized”* with *“sociodemographic.”*
- e. We assume that the reviewer meant to refer to Figure 5, where, indeed, we show relative and not absolute frequencies. We updated the legend and the caption to reflect that.
- f. We experimented with merging the two figures into a single one as suggested by the reviewer, but we found the result to be visually more confusing and harder to read. Therefore, we decided to keep distinct for now the two figures.

Responses to Reviewers' Comments for Manuscript

NATHUMBEHAV-24051824A

On the Conversational Persuasiveness of Large Language Models

Addressed Comments for Publication to

Nature Human Behaviour

by

Francesco Salvi, Manoel Horta Ribeiro, Riccardo Gallotti, Robert West

Authors' Response to Reviewer 1

General Comments. In this second-round review of “On the Conversational Persuasiveness of Large Language Models: A Randomized Controlled Trial,” the authors provided new citations, analyses, and conclusions in support of their findings that gen-AI personalized persuasion can outperform human persuasion in a debate context. Overall, this review will be short, because I thought they adequately tested the additional effects and mechanisms I suggested, and they provided the appropriate language to address some of the limitations of the current work. Like in the first round, I found this to be a very impressive paper, and I think it will make a noteworthy contribution to the literature. I only have a few last final suggestions.

Response: We thank the reviewer for their kind words and appreciation.

Comment 1

It would be nice to have a couple examples of what sociodemographic information was provided to the personalized conditions. It doesn't need to be exhaustive but given the ongoing debate about what kinds of variables are most effective in generating personalized persuasion, it would be nice to have a parenthetical or even a sentence providing a little more context on that. For example, I had originally thought the sociodemographic information was simply age, gender, and race; however, in SI-5, I learned that it included political identification. In short, I think it would help to advance the broader discussion on personalized persuasion if the authors mentioned in the primary text a couple examples of that sociodemographic information, how many pieces of information there was in total, and whether the humans/AI had access to all of that information or simply a random subset of it. In this vein, it might be worth mentioning in the General Discussion how the effects for AI-personalized might have been even stronger if it had been provided with even more meaningful psychological variables (e.g., opponents' moral bases).

Response: We agree with the reviewer about the importance of emphasizing what kind of sociodemographic information was provided in the personalized condition. In our new revision, we explicitly mention the full list of provided pieces of information ("Gender, Age, Ethnicity, Education Level, Employment status, and Political affiliation") in four points in the main text:

- In the caption of Fig. 1.
- In lines 118-120, when discussing the pre-treatment survey that all participants had to complete.
- In lines 181-182, when explaining treatment conditions.
- In lines 452-453, when discussing the implications of our results and their potential impact in online environments.

Additionally, we incorporated the reviewer's suggestion in the relevant paragraph (lines 450-458) of our Discussion, which now reads:

"We emphasize that the effect of personalization is particularly remarkable given how little personal information was collected (Gender, Age, Ethnicity, Education Level, Employment status, and Political affiliation) and despite the extreme simplicity of the prompt instructing LLMs to incorporate such information (see Supplementary Information 2.5 for the complete prompts). Even stronger effects could thus be obtained by exploiting individual psychological attributes, such as personality traits and moral bases, or by developing stronger prompts through prompt engineering, fine-tuning, or specific domain expertise."

Comment 2

I really appreciated the discussion in the General Discussion about the study being a “proof of concept” with real world corollaries. I thought a single sentence to this effect in the introduction could be valuable, too (e.g., “Although this study uses a carefully controlled debate paradigm, it nonetheless serves as a meaningful proof of concept for the form in which debates like these take place online, such as through synchronous discussions in comment threads on websites like Facebook and Reddit.”). In this vein, I thought it could be important to mention in the General Discussion that the structure of the experiment might have put a premium on information that was “fact-based and logical,” as this is how formal debates are commonly determined to have winners and losers. In contrast, when “debating” casually with a friend, research has found that using I-statements and personal narratives can be more effective. That is to say that the AI advantage might have been facilitated by logic-favoring context of the experimental design. At the same time, that is not to say that if an AI had been prompted to use personal narratives, it couldn’t have outperformed humans in a more casual form of political discourse, too.

Again, overall, I found this paper to be very well written and comprehensive and think it will make an excellent contribution to both the work on gen-AI as well as personalized persuasion!

Response: We incorporated a slightly rephrased version of the sentence suggested by the reviewer at the end of our Introduction:

"While our study uses a structured debate format, it nonetheless serves as a valuable proof of concept for how similar debates occur online, such as in synchronous discussions on platforms like Facebook and Reddit."

While we agree with the reviewer that, in principle, the structure of the experiment might have put a premium on information that was "fact-based and logical", we found no practical evidence of *logicality* having a meaningful effect in altering participant’s beliefs.

As we reported in our previous response, when controlling for the degree of logicity, we find that the *Analytic* feature from LIWC increases the odds of higher post-treatment agreement by only +1.4% ($p = 0.97$) while leaving virtually unchanged all the treatment effects

Authors' Response to Reviewer 2

General Comments. I thank the authors for successfully addressing all the points I raised in the round of review. Great job and precise and commendable review process.

Response: We thank the reviewer for their kind words.

Authors' Response to Reviewer 3

General Comments. Thank you very much for addressing all my suggestions and adding additional explanations both in the cover letter and the main manuscript.

Your response to my comment #5 (pull the personalization prompt to the main manuscript) is very helpful. It made me think that you could even play this up a little in the discussion – only if you want, not necessary at all. What I mean is that the prompt is incredibly simple, which means that the effects you observe are based on minimal prompting (and prompt engineering presumably). Might be worth highlighting.

I have nothing else to add, congratulations on a great paper!

Response: We thank the reviewer for their kind words. Following the reviewer's recommendation, we slightly rewrote our discussion about the effect of personalization (lines 450-458), which now reads:

We emphasize that the effect of personalization is particularly remarkable given how little personal information was collected (Gender, Age, Ethnicity, Education Level, Employment status, and Political affiliation) and despite the extreme simplicity of the prompt instructing LLMs to incorporate such information (see Supplementary Information 2.5 for the complete prompts). Even stronger effects could thus be obtained by exploiting individual psychological attributes, such as personality traits and moral bases, or by developing stronger prompts through prompt engineering, fine-tuning, or specific domain expertise.